# Transient and general synthesis of high-density and ultrasmall nanoparticles on two-dimensional porous carbon via coordinated carbothermal shock

Wenhui Shi[1], Zezhou Li[2], Zhihao Gong[3], Zihui Liang[1], Hanwen Liu[1], Ye-Chuang Han[4], Huiting Niu[5], Bo Song[1], Xiaodong Chi [1], Jihan Zhou[2], Hua Wang[3], Bao Yu Xia [5] ✉, Yonggang Yao [1] ✉ & Zhong-Qun Tian [4] ✉

Carbon-supported nanoparticles are indispensable to enabling new energy technologies such as metal-air batteries and catalytic water splitting. However, achieving ultrasmall and high-density nanoparticles (optimal catalysts) faces fundamental challenges of their strong tendency toward coarsening and agglomeration. Herein, we report a general and efficient synthesis of high-density and ultrasmall nanoparticles uniformly dispersed on two-dimensional porous carbon. This is achieved through direct carbothermal shock pyrolysis of metal-ligand precursors in just ~100 ms, the fastest among reported syntheses. Our results show that the in situ metal-ligand coordination (e.g., $N \rightarrow Co^{2+}$) and local ordering during millisecond-scale pyrolysis play a crucial role in kinetically dominated fabrication and stabilization of high-density nanoparticles on two-dimensional porous carbon films. The as-obtained samples exhibit excellent activity and stability as bifunctional catalysts in oxygen redox reactions. Considering the huge flexibility in coordinated pre-cursors design, diversified single and multielement nanoparticles (M = Fe, Co, Ni, Cu, Cr, Mn, Ag, etc) were generally fabricated, even in systems well beyond traditional crystalline coordination chemistry. Our method allows for the transient and general synthesis of well-dispersed nanoparticles with great simplicity and versatility for various application schemes.

Ultrasmall nanoparticles (NPs) have a wide range applications in catalysis, biomedicine, and energy conversion due to their size-dependent unique properties[1–3]. In catalysis, NPs are anchored on porous carbon or oxide supports to prevent agglomeration. Their performance is highly dependent on their size and loading[4–6], smaller size and higher loading usually result in more low-coordinated atoms and active sites, significantly improving reaction activity and kinetics[7,8]. However, synthesizing ultrasmall and high-density NPs is challenging due to their thermodynamic instability and tendency to aggregate, particularly at high temperatures[9–11]. Wet chemistry methods (e.g., colloidal synthesis) can produce ultrafine and even monodispersed NPs, but their complex synthetic routes, surfactant coverage, and weak catalyst-support interactions limit their wide utilization in practical applications[12–15].

Different from wet chemistry methods, direct pyrolysis of organic substances loaded with metal precursors is a common strategy to synthesize carbon-supported NPs[16–19]. Recently, carbothermal shock (CTS) technology has emerged as a promising methods for synthesizing a wide range of catalysts (size and composition) due to its

precise heating control and extreme efficiency (in seconds)[4,20–22]. However, particle agglomeration and sintering are inevitable at high temperatures, typically resulting in larger nanoparticles (>10 nm, for non-noble metals). To enable the synthesis of ultrasmall NPs, enhanced metal-substrate interactions have been explored by introducing highly defective substrates or heteroatom doping (e.g., S, N, P) as anchoring sites for effective dispersion and/or high surface coverage[23–25]. However, these methods often require special substrate treatments or strict bond formation, while the formed NPs are only anchored on the backbone substrate, leading to relatively complex syntheses and/or low surface dispersion. A comparison of various CTS methods is summarized in Supplementary Table 1.

Alternatively, metal–organic frameworks (MOFs) possess atomically dispersed metal nodes linked by organic ligands, acting as an ideal platform to synthesize high surface area carbon-supported catalysts through pyrolysis (Supplementary Fig. 1)[26–33]. However, due to the high-density metal nodes, traditional hours of high-temperature pyrolysis can easily result in serious particle agglomeration (sometimes >100 nm)[34–36]. Previously, rapid CTS pyrolysis was used for MOF microcubes to synthesize ultrasmall NPs (~4 nm)[33], yet most NPs were found to be buried inside the carbon matrix derived from pyrolyzing MOF ligands, thus causing serious metal deactivation and substantial resistance in mass transfer (electrolyte and gas) along with inferior catalytic performances. To mitigate these problems, special MOF chemistry designs (e.g., using volatile Zn) or posttreatments (e.g., acid etching) are adopted to obtain ultrasmall NPs or even single-atom catalysts[37,38]. In addition, sacrificial

templates are often used to increase particle exposure and structure openness[39,40]. However, the synthesis of MOFs involves fine chemistry design and is a time- and energy-consuming process, resulting in low scalability and limited types of catalysts based on MOF species. In addition, in the subsequent pyrolysis of crystalline MOFs, it is still challenging to avoid MOF structure collapse and particle aggregation. Therefore, a more general and controllable method is highly desirable to achieve not only ultrasmall and high-density NPs but also open porous structures that are essential for catalyst exposure and efficient mass transport, thus significantly improving the versatility of catalyst synthesis and high-performance catalysis.

Herein, we report the efficient and general synthesis of high-density and ultrasmall NPs uniformly dispersed and anchored on two-dimensional porous carbon, achieved by directly coordinated carbothermal shock (coordinated CTS, -100 ms) pyrolysis of the metal-ligand precursors without the need for time-consuming MOF preparation (Fig. 1a). In the process, the metal-ligand precursor (e.g., $Co^{2+}$ and dimethylimidazole $C_5H_8N_2$) undergoes in situ assembly into a more ordered structure with strong coordination ($N \rightarrow Co^{2+}$) at low temperature, which is then explosively carbonized at high temperature to synthesize uniformly dispersed Co NPs stabilized by N-doped 2D porous carbon. Meanwhile, as a result of rapid heating ($10^4$ °C/s), 2D thin and porous carbon films are formed with exposed NPs; the porous structure with a multitude of open pores is essential to exposing active sites and enabling electrolyte penetration for improved catalytic performances. We compared the pyrolysis time,

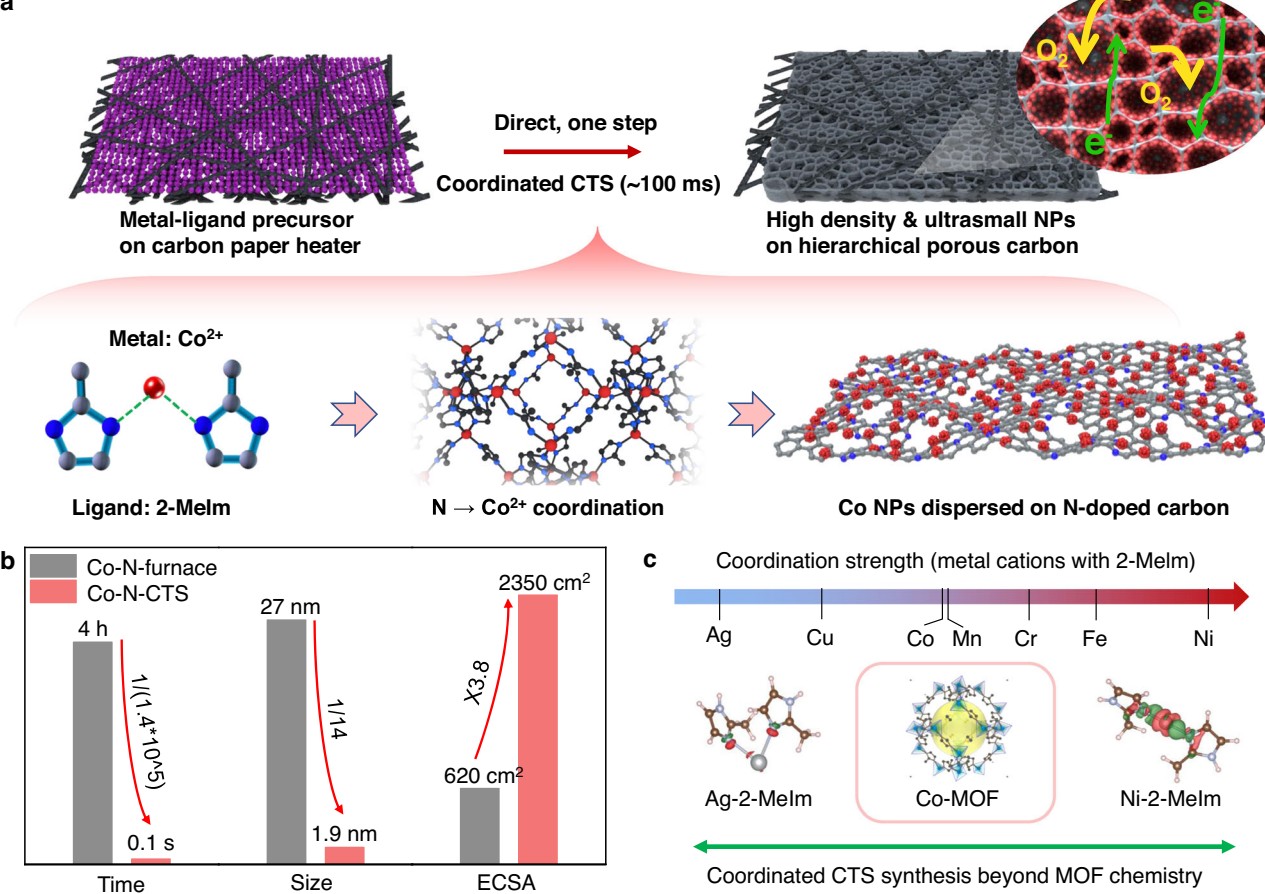

**Fig. 1 | The direct synthesis of high-density, ultrasmall NPs via coordinated CTS.**
**a** Coordinated carbothermal shock (CTS): the whole synthesis process takes less than 1 s by direct shock pyrolysis of the coordinated metal-ligand precursor (Co-dimethylimidazole (Co-2-MeIm)), which leads to high-density, ultrasmall nanoparticles (NPs) dispersed and stabilized on two-dimensional porous carbon. **b** Compared to furnace annealing, our method shows a time reduction by at least -5 orders of magnitude, a size reduction from 27 to 1.9 nm, and an increase in electrochemical surface area (ECSA) of 3.8 times. **c** The general synthesis of ultrasmall nanoparticles by coordinated CTS beyond delicate metal-organic framework (MOF) chemistry.

size, and surface area of samples synthesized by our method (denoted as Co-N-CTS) and conventional furnace pyrolysis (denoted as Co-N-furnace), where our method showed a time reduction by at least ~5 orders of magnitude, a size reduction from 27 to 1.9 nm, and an increase in electrochemical active surface area (ECSA) of 3.8 times (Fig. 1b). Importantly, our method is universal for synthesizing various single metal and multielement alloy NPs, even in systems well beyond MOF chemistry (Fig. 1c), demonstrating the extreme simplicity and versatility of shock pyrolysis combined with coordination chemistry toward well-dispersed and exposed NPs.

## Results

### Material synthesis and structural characterizations

The synthesis process simply involves dropping the metal-ligand precursor onto carbon paper and then initiating the high-temperature CTS process to obtain well-dispersed NPs on 2D porous carbon (Supplementary Fig. 2). The carbon paper behaves as both the precursor holding substrate and the CTS platform, as it has suitable conductivity as a catalyst substrate and for rapid Joule heating. In our synthesis, the precursor was composed of metal salt and ligands, and we used $Co(NO_3)_2$ and dimethylimidazole ($C_5H_8N_2$, 2-MeIm) for demonstration, which can form coordinated polymers and various MOFs, such as ZIF-67[39,41,42]. The $Co(NO_3)_2$ ethanol solution turns blue after adding an equal volume of 2-MeIm ethanol solution, indicating that the Co ions coordinate with 2-MeIm to form a metal-organic complex (MOC) (Supplementary Fig. 3). The carbon paper loaded with MOC was then Joule heated to a high temperature to induce CTS synthesis, where the peak temperature was 800–1200 °C and lasted for ~100 ms (Supplementary Fig. 4)[22,43–45].

From the SEM image, we can observe porous films attached to the carbon paper substrate, where the films show a multitude of pores at the micro- and nanoscale (Fig. 2a and Supplementary Fig. 5). These carbon films are the pyrolyzed products of metal-ligand precursors, and the multitude of porosities is a result of rapid heating. The SEM image exhibits a pore size distribution from 1 to 10 μm, with an average micropore size of ~0.6 μm. Optical microscopy shows a 2D thin film with a lateral size of 185.8 μm (Supplementary Fig. 6). Atomic force microscopy (AFM) was used to characterize the 2D porous carbon film (Fig. 2b), where thin 2D porous films decorated with small NPs were observed. These carbon films have a thickness of ~3.76 nm, as determined by edge scanning analysis (Supplementary Fig. 7). The porous structure and distribution were further characterized by nitrogen adsorption and desorption, where Co-N-CTS demonstrated a higher Brunauer–Emmett–Teller (BET) specific surface area of 326 m²/g than the Co-N-furnace sample (190 m²/g) (Fig. 2c). Moreover, Co-N-CTS shows multiple pore size distributions between 1 and 10 nm at the nanoscale (Supplementary Fig. 8).

During the rapid CTS process (with a heating rate of $10^4$ °C/s), the decomposition of metal precursors and ligands will instantly generate a large amount of vigorously diffused gases that act as pore-forming agents, which in turn lead to the generation of porous structures with thin 2D porous carbon instead of dense and compact structures. We therefore studied the heating rate effect (from ~0.1 to $10^4$ °C/s, Supplementary Fig. 9 and Supplementary Table 2), which is believed to be critical to creating two-dimensional porous structures and thin carbon supports, as revealed by SEM, BET, and electrochemically active surface area (ECSA, Supplementary Figs. 8 and 10–12). A compact and dense structure was obtained with slow heating (0.1–100 °C/s), while increasingly open and porous structures were obtained with an increased heating rate. However, only at a sufficiently high heating rate, e.g., $10^4$ °C/s, do we obtain an open porous structure that is ideal for nanoparticle exposure, electrolyte penetration, and mass/gas transportation (Supplementary Fig. 13). The critical heating rate dependence of pore formation is further confirmed in Cu with a benzenetricarboxylic acid ligand (Cu-BTC)

precursor (Supplementary Figs. 14–17). Meanwhile, the pyrolysis temperature will also induce changes in the porous structure and nanoparticle size (Supplementary Fig. 18), crystallinity (Supplementary Fig. 19a), and graphitization (Supplementary Fig. 19b). Previously, researchers used different methods to artificially create open pores[16,29,32], but now the porous structure can be easily formed because of rapid heating and explosive carbonization, demonstrating the advantages of our coordinated CTS pyrolysis.

Figure 2d and Supplementary Fig. 20 show a low magnification TEM image of the synthesized Co-N-CTS sample with a two-dimensional porous structure, where ultrasmall and well-dispersed NPs are observed (~3.2 nm). The high-resolution TEM confirms the high-density distribution of Co NPs, showing a lattice fringe spacing of 0.21 nm, which can be assigned to the metallic Co (111) crystal plane (Supplementary Fig. 21). Surprisingly, such a homogeneous distribution of NPs can be synthesized within 100 ms. Moreover, high-angle annular dark-field scanning TEM (HAADF-STEM) and corresponding EDS mapping of the Co-N-CTS sample are also investigated. As shown in Fig. 2e and Supplementary Fig. 22, Co is mainly distributed in the nanoparticle; N is distributed in not only the Co nanoparticle but also the carbon region, indicating the formation of N-doped carbon from the pyrolysis of the 2-MeIm ligand ($C_5H_8N_2$). Particularly, from the overlap of Co-N mapping, it is clear that Co and N are spatially correlated, indicating that the Co NPs are dispersed and stabilized by the N-doped carbon, originating from the strong metal-substrate (Co-N) interaction, as will be discussed later.

To understand the importance of Co-N coordination and rapid CTS pyrolysis, we performed two control samples: (1) CTS pyrolysis of the Co salt precursor without the 2-MeIm ligand (denoted as Co-CTS) and (2) furnace treatment of the MOF (i.e., Co-N-furnace). Large particles (~140 nm) and agglomerated particles were observed after pyrolyzing the Co precursor without 2-MeIm coordination (Supplementary Fig. 23), largely because Co atoms are not confined and can freely move to aggregate at high temperatures. On the other hand, dense and compact structures can be observed after conventional furnace treatment (750 °C, 4 h), where serious particle aggregation was observed (~27 nm) (Supplementary Figs. 24–27), which may be due to the slow and long-term high-temperature heating that leads to the collapse and condensation of pyrolyzed products and long-range transport of atoms toward aggregation. These results indicate that both coordination and ultrafast pyrolysis are essential for the synthesis of ultrasmall, well-dispersed, and stabilized NPs.

Another unique advantage of the CTS process is the high N retention after rapid pyrolysis - up to 24 at%, which is in sharp contrast to conventional pyrolysis with typical N content <10%[26,27]. The high N content is critical to ensure that high-density Co NPs are uniformly dispersed and stabilized by forming Co-N bonds, as shown in the X-ray photoelectron spectroscopy (XPS) measurement (Fig. 2f and Supplementary Fig. 28 and Table 3). We especially focused on N peak deconvolution, which shows not only pyridinic, pyrrolic, graphitic, and oxidized N but also Co-N peaks[46,47]. The relative contents of Co-N decrease at a higher pyrolysis temperature, which may explain the increasing size of Co NPs due to less N for anchoring and stabilization. We displayed the relationship between the N content and particle size, where an inverse correlation is found, supporting our explanation that Co-N content is critical for good dispersion (Supplementary Fig. 28d). In addition, by detailed analysis of the Co XPS spectra (Supplementary Fig. 29 and Tables 4 and 5), a strong $Co^0$ signal was detected at 778.5 eV, while weak $Co^{2+}$ and $Co^{3+}$ signals were also observed, suggesting surface oxidation of Co NPs[48,49]. Additionally, the peak position of metallic Co 2p shifts to a higher binding energy (~0.3 eV) in our Co-N-CTS sample, indicating partial electron transfer from Co to N and effective modulation of the electronic structure in Co-N-CTS by the strong metal-substrate interaction between Co NPs and N-doped carbon[50]. The metal-substrate interaction and electron transfer are

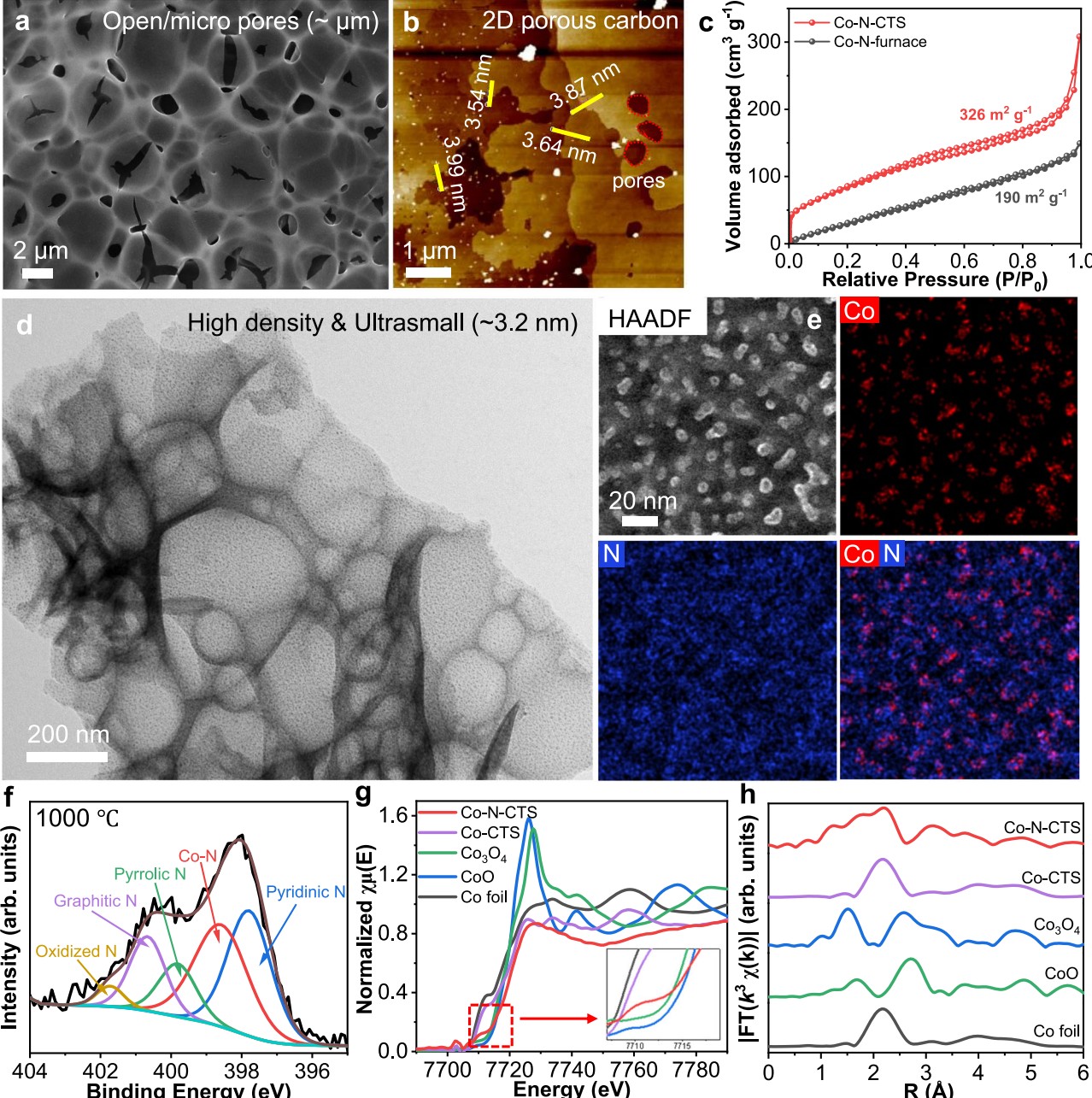

**Fig. 2 | Coordinated CTS synthesized high-density and ultrasmall nanoparticles on two-dimensional porous carbon. a** The porous structure of Co-N-CTS with a multitude of open pores. **b** AFM image of 2D porous carbon showing a thin film structure and a thickness of <4 nm. **c** N₂ sorption isotherms of Co-N-CTS and Co-N-furnace. **d** TEM image of Co-N-CTS with a size of ~3.2 nm. **e** HAADF and EDS mapping images of Co-N-CTS. **f** High-resolution XPS deconvolution of the N element of Co-N-CTS-1000. **g** The normalized XANES spectra and **h** $k^3$-weighted Fourier transform of EXAFS spectra at the Co $K$-edge of Co-N-CTS, Co-CTS, and reference samples.

further verified by charge density distribution calculations of Co NP on carbon and on N-doped carbon (Supplementary Fig. 30), where Co on N-doped carbon loses more electrons (−0.483 e⁻), verifying the electron transfer from Co to N atoms.

To further investigate the electronic structure and coordination environment of Co species in Co-N-CTS, Co $K$-edge X-ray absorption near-edge structure (XANES) spectroscopy and extended X-ray absorption fine structure (EXAFS) analysis were also conducted (Fig. 2g, h)[51,52]. For comparison, Co foil, CoO, Co₃O₄, and Co-CTS were used as the reference samples. The Co $K$-edge in the XANES spectra shows that the absorption near edge spectrum for Co-N-CTS lies between those for Co foil and CoO (Fig. 2g), indicating partial electron transfer from Co to N with a valence state between 0 and +2. However,

the absorption near the edge spectrum of Co-CTS lies near that for Co foil and shifts slightly to the right, indicating the coexistence of metallic Co and slight surface oxidized Co. Additionally, the Fourier transformed EXAFS (FT-EXAFS) spectra of the Co $K$-edge for Co-N-CTS further indicate the coexistence of Co-N/O and metallic Co-Co scattering paths (Fig. 2h), whereas for Co-CTS, only a predominant peak at ≈2.16 Å and a weak peaks at ≈1.48 Å were observed, which is typically assigned to the Co-Co and Co-O coordination, respectively[53–55].

## Electrochemical performances

To demonstrate utility, the synthesized Co-N-CTS catalysts are used as bifunctional catalysts for the Zn-air cathode. First, we screened the catalytic activity of Co catalysts pyrolyzed at different temperatures of

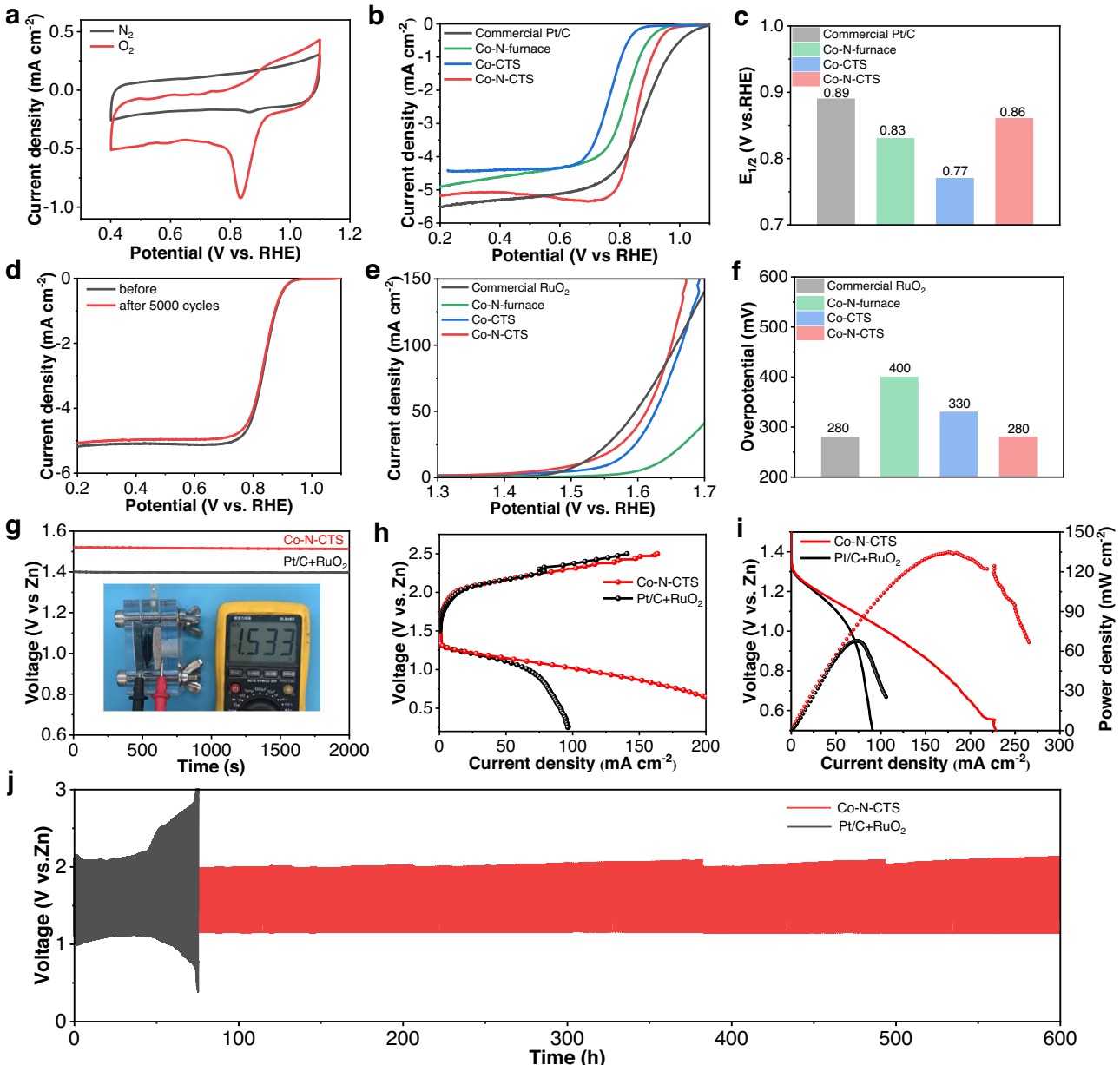

**Fig. 3 | OER, ORR, and Zn-air battery performance of our catalysts. a** CV curves of Co-N-CTS in 0.1 M KOH solution with saturated $N_2$ and $O_2$. **b** The ORR LSV curves of Co-N-CTS and control samples. **c** The half-wave potential of Co-N-CTS and control samples. **d** The ORR-LSV curves before and after 5000 cycles. **e** The OER LSV curves of Co-N-CTS and control samples. **f** The overpotential of Co-N-CTS and control samples at 10 mA/cm². **g** Open voltage of the Zn-air battery. **h** The charge and discharge curves of Co-N-CTS and Pt/C + RuO₂. **i** The corresponding power density curves. **j** The duration test of the Zn-air battery.

600, 800, 1000, and 1200 °C (denoted as Co-N-CTS-600, Co-N-CTS-800, Co-N-CTS-1000, and Co-N-CTS-1200) and found that Co-N-CTS-1000 shows the best OER and ORR activity (Supplementary Fig. 31). Thus, we define Co-N-CTS-1000 as Co-N-CTS for the catalytic study. For the oxygen reduction reaction (ORR), the electrochemical catalytic activity of Co-N-CTS was tested by cyclic voltammetry (CV) over the potential range of 0.4 to 1.1 V versus reversible hydrogen electrode (RHE) at room temperature in $N_2/O_2$-saturated 0.1 M KOH electrolyte (Fig. 3a). A peak appearing at ~0.85 V versus RHE is observed in the O₂-saturated electrolyte, indicating the good ORR activity of Co-N-CTS. Linear sweep voltammetry (LSV) curves and the half-wave potential of different samples were collected to investigate the ORR activity (Fig. 3b, c). To demonstrate the stability, accelerated potential cycling tests were conducted in the interval between 0.6 and 1.0 V (versus RHE) at a scanning rate of 50 mV/s for up to 5000 cycles. As displayed in Fig. 3d,

Co-N-CTS-1000 shows a negative shift of only 6 mV for the half-wave potential, indicating good ORR stability in accelerated electrochemical cycling. In addition, after the current−time (*i*−*t*) chronoamperometry measurements at a constant potential of 0.6 V versus RHE for 10 h, the retention rate of current density for the Co-N-CTS catalyst is 94.8%, higher than that of commercial Pt/C (77.4%) (Supplementary Fig. 32), further verifying the excellent ORR stability of the Co-N-CTS catalyst.

In terms of the oxygen evolution reaction (OER), Co-N-CTS also displays excellent OER performance even compared to noble RuO₂ catalysts (Fig. 3e). The overpotential of Co-N-CTS and RuO₂ is ~280 mV, which is similar to that of commercial RuO₂ but much lower than that of the other control samples (Co-CTS or Co-N-furnace, Fig. 3f). Additionally, Co-N-CTS displays a much larger current density in the high potential region, which can be attributed to the unique porous structure facilitating mass transport. As shown in Supplementary Fig. 33, the

Tafel slope of Co-N-CTS (94.5 mV/dec) is slightly larger than that of commercial $RuO_2$ (77.5 mV/dec) and lower than those of Co-CTS (96.8 mV/dec) and Co-N-furnace (160.3 mV/dec), suggesting the fast OER kinetics of Co-N-CTS. The intrinsic activities of Co-N-CTS are also investigated by normalizing the current based on the ECSA, where the ECSA-normalized specific activity of Co-N-CTS is still higher than that of Co-N-furnace, indicating higher intrinsic activity in Co-N-CTS samples (Supplementary Fig. 34). The long-term durability of the OER was also tested at a constant current density of 10 mA/cm$^2$ (chronopotential test) for 20 h (Supplementary Fig. 35). The Co-N-CTS catalyst maintained its initial potential after 20 h and surpassed that of the noble $RuO_2$ catalyst, indicating excellent OER stability.

We also performed poststructure analysis for Co-N-CTS catalysts after the long-term stability test. The Co-N-CTS catalysts still maintained the open and porous structure with high-density Co NPs uniformly dispersed on the 2D porous carbon, and no obvious agglomeration was found (Supplementary Fig. 36). Additionally, the XRD and XPS results showed a metallic Co phase with an increasing oxidized surface after the stability test of ORR and OER (Supplementary Fig. 37), which is commonly observed for nonnoble metals[56–58]. These results validate the good structural and performance stability of the Co-N-CTS catalyst.

The ORR and OER activity of our catalyst is comparable or superior to that of most of the reported state-of-the-art nonprecious metal catalysts (Supplementary Fig. 38 and Table 6), which could be attributed to the following reasons. For one, the high-density and ultrasmall Co NPs with more surface and low-coordinated atoms afford abundant active sites to promote and increase the apparent catalytic activity[59,60]. In addition, the strong metal-substrate interaction between Co NPs and N-doped carbon and their electron transfer significantly regulate the electronic structure of Co to optimize the binding energies toward intrinsically enhanced catalytic activity[41,61]. Last but not least, the catalysts are dispersed on open and porous structures with large surface areas, which effectively exposes these active sites, facilitates electrolyte penetration, and benefits gas/mass transportation, all of which are beneficial for electrocatalytic ORR and OER, especially at large current densities.

The above results indicate the good ORR and OER catalytic activity of Co-N-CTS, which can be used as a bifunctional catalyst for the Zn-air battery cathode. With a loading of 1 mg/cm$^2$, the assembled battery affords an open-circuit voltage of 1.533 V and remains unchanged after 2000 s (Fig. 3g). The charging/discharging curves and power density profiles are displayed in Fig. 3h. The Co-based battery shows an even smaller voltage gap than that of the Pt/C + $RuO_2$-based battery during the charging/discharging process, illustrating an exceptional rechargeable capability using Co-N-CTS catalysts. The Co-based battery also delivers a power density of 140 mW/cm$^2$, which is significantly higher than 60 mW/cm$^2$ for the Pt/C + $RuO_2$-based battery (Fig. 3i). The long-term rechargeability of the Co-based battery is also evaluated by continuous galvanostatic charging/discharging at a current density of 10 mA/cm$^2$ with 60 min per cycle (discharge 30 min and charge 30 min) (Fig. 3j), which can operate for a long time of 600 cycles in 600 h. However, Pt/C + $RuO_2$ completely failed after 75 h of operation.

## Synthesis mechanism analysis during coordinated-CTS

Our work is to study the effect of organic ligands and coordination chemistry during the rapid CTS process. Therefore, it is necessary to understand the process and mechanism of obtaining high-density and ultrasmall NPs. The reaction between $Co(NO_3)_2$ and 2-MeIm was tracked by thermogravimetric analysis-derivative thermogravimetry technology (TGA-DTG). As shown in Fig. 4a, there are major steps during the pyrolysis process: (1) the MOC precursor structure transformation and decomposition with a weight loss of 45.3 % at ~300 °C, (2) the carbonization process and weight loss of 21 % between 300 and 700 °C, and (3) the graphitization process at higher temperatures

(>700 °C)[62,63]. Notably, there are three endothermic peaks in the DTG profile below 300 °C, which are important clues to identifying the structural transformation of the MOC precursor at low temperature. TGA-DTG profiles of the control samples ($Co(NO_3)_2$ and 2-MeIm separately) are also measured, as shown in Supplementary Fig. 39. There is a weight loss of 100 % (i.e., volatized) at >230 °C in 2-MeIm heating, indicating that the strong interaction between $Co(NO_3)_2$ and 2-MeIm can effectively reduce volatilization and evaporation during pyrolysis[64,65].

To understand the phase transformation at low temperature, ex situ FT-IR technology is applied to detect species evolution at three endothermic peaks below 300 °C. As shown in Fig. 4b, the MOC precursors heated to different temperatures through CTS were characterized by FT-IR. The MOC precursor shows vibration peaks of -CH at 1380 cm$^{-1}$, which belongs to the 2-MeIm molecule. For MOC heated to 100 and 180 °C, the vibration peak of Co-N at 400 cm$^{-1}$ starts to appear and turns strong, indicating that the Co-N coordination in the MOC precursor increases. Notably, there is an obvious vibration peak of -OH at 3500 cm$^{-1}$ appearing at 180 °C, which can be attributed to the adsorbed water from the ambient environment into the MOC sample before the FT-IR test. The adsorption of water is a strong indication for the formation of a porous structure or ordered framework in MOC heated to 180 °C. At an even higher temperature (270 °C), the vibration peaks of -OH and Co-N become very weak due to MOC precursor decomposition[66,67].

X-ray diffraction (XRD) measurements revealed the structure information before (MOC) and after the CTS process (Co-CTS). Notably, the XRD pattern of MOC is different from that of either $Co(NO_3)_2$ or 2-MeIm, indicating that the Co ions have already coordinated with 2-MeIm. Meanwhile, it is also largely different from the ordered MOF structure (ZIF-67). This is particularly obvious when looking at the crystalline peak (011) in ZIF-67, where such a peak is absent in our MOC precursor. In contrast, the XRD profile of our MOC precursor shows multiple small peaks but without sharp and well-defined crystalline peaks, indicating that only short-range or local ordering exists in our MOC precursor (Supplementary Fig. 40)[68,69]. After the CTS process, the peaks of the MOC precursor disappeared, and carbon peaks appeared, suggesting that the precursors decomposed and further carbonized.

To further characterize the structural transformation of the MOC precursor, in situ grazing incidence wide-angle X-ray spectrum (GIWAX) technology was carried out. Figure 4c, d show the in situ heating GIWAX results of MOC (Co and 2-MeIm) at different temperatures and using Cu-2-MeIm as a comparison. The GIWAX mapping profile from room temperature (30 °C) to high temperature (300 °C) can be divided into three stages that correspond well to the three endothermic peaks in the DTG profile. At 70 °C, the main peak at 9 nm$^{-1}$ shifts to the right, indicating that the lattice distance becomes large, which can be attributed to the increase in the Co-N coordination bond. The structural transformation occurs at 140 °C, where multiple crystalline peaks are formed, resembling the formation of the locally ordered metal-organic complex. This corresponds to the formation of the porous structure revealed by FT-IR measured at ~180 °C, indicating that the structural transformation is mainly excited by temperature and is not affected by heating rate. When the temperature rises to 190 °C, these crystalline peaks disappear, suggesting that the ordered MOC precursor starts to decompose. The corresponding diffraction ring also clearly revealed the structural transformation process during in situ heating (Supplementary Fig. 41). In contrast, there is no obvious structural transformation for the Cu and 2-MeIm precursors during heating treatment (Fig. 4d), particularly in the absence of multipeak ordered structures. The corresponding diffraction pattern also confirms no structural transformation with increasing temperature (Supplementary Fig. 42). As a result, aggregated particles are observed in the Cu-2-MeIm samples (Supplementary Fig. 43). These

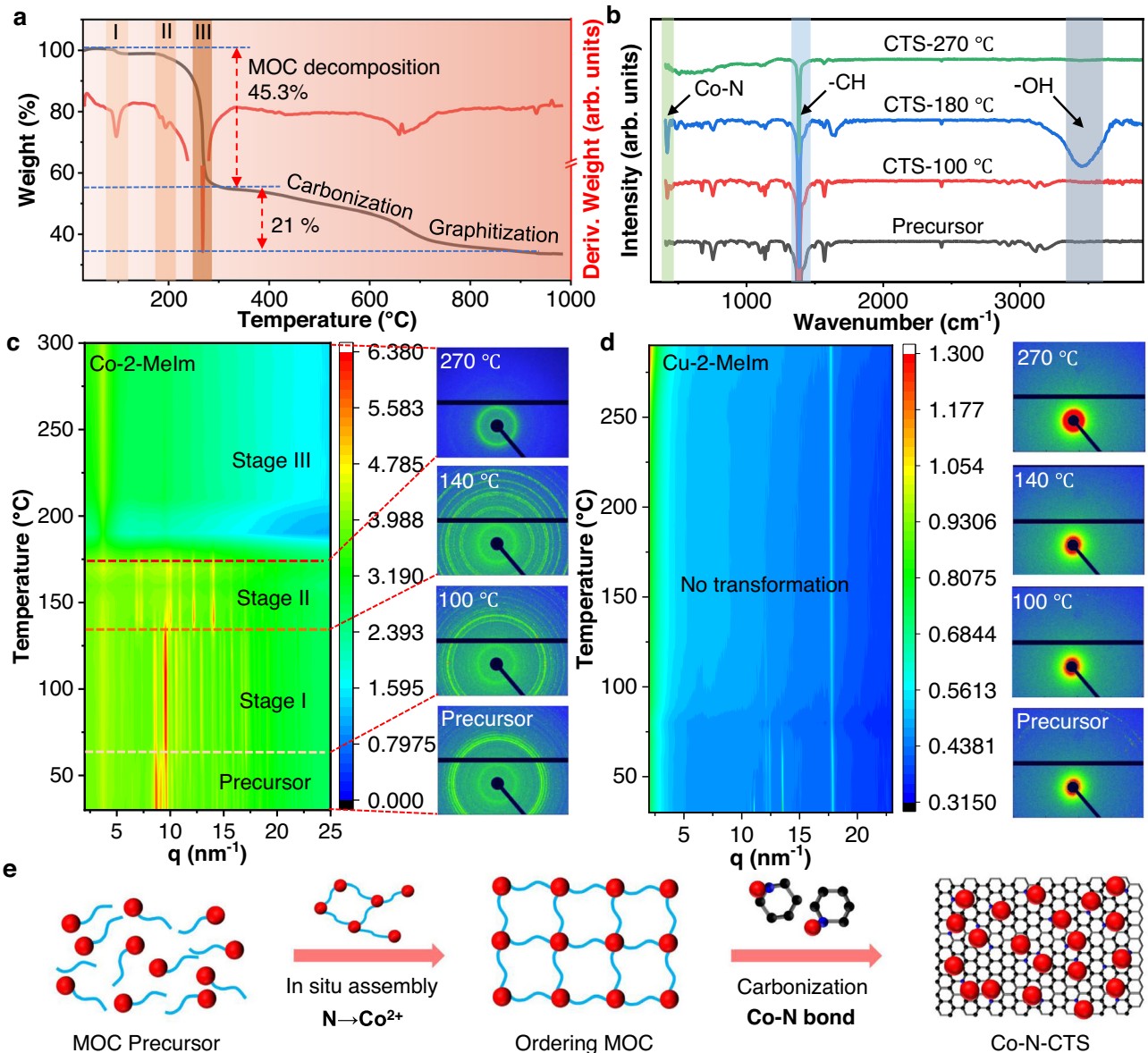

**Fig. 4 | Synthesis process and structural evolution in coordinated CTS pyrolysis. a** The TGA-DTG profile of metal-organic complex (MOC) from 50 to 1000 °C in Ar gas. **b** FT-IR spectra of MOC-derived samples after CTS treatment at different temperatures. **c** The in situ GIWAX spectrum of Co-dimethylimidazole (Co-2-MeIm) from 30 to 300 °C and the corresponding diffraction pattern at the transformation temperature. **d** The in situ GIWAX spectrum of Cu-dimethylimidazole (Cu-2-MeIm) (control) from 30 to 300 °C and the corresponding diffraction pattern at characteristic temperatures. **e** Schematic process of structure evolution in coordinated CTS pyrolysis.

results indicate that the in situ assembly process is essential for the synthesis of high-density and ultrasmall NPs.

Based on the above characterization and observation, we illustrated the synthesis process by CTS pyrolysis, as shown in Fig. 4e. First, the MOC (metal and ligand) precursor demonstrates a partially and loosely coordinated structure. As the temperature increases, the MOC precursor starts to form locally ordered framework structures (ordered MOC) through in situ assembly, which is confirmed by the FT-IR and GIWAX results. Finally, when the temperature further increases, the MOC precursor starts to decompose, followed by later carbonization and graphitization. Note that the in situ assembly to form a coordinated structure and connected network is crucial to forming well-dispersed NPs stabilized on 2D porous N-doped carbon. Without these coordination bonds, both metal and ligands will decompose and evaporate rapidly, leaving large aggregates behind. To differentiate

conventional CTS (without ligand coordination), we refer to our CTS pyrolysis of the metal-ligand precursor as coordinated CTS pyrolysis.

**General synthesis of single and multielement alloy NPs**
Based on the above understanding, a general synthesis of high-density and ultrasmall NPs is demonstrated in Fig. 5. Since the coordination effect between the metal salt and ligand is important to induce in situ assembly and later ultrasmall NPs formation, we used density functional theory (DFT) to calculate the formation energy (coordination) between different metal ions and ligands (2-MeIm) to guide our synthesis (Fig. 5a and Supplementary Fig. 44 and Table 7)[70–76]. First, the formation energy for the model catalyst Co-2-MeIm is −3.44 eV, which leads to excellent size and dispersion (Fig. 5c). Similar or even stronger binding energy was found in Mn, Cr, and Ni coordinated with 2-MeIm (−3.45, −4.14, and −5.8 eV) and accordingly uniform nanoparticle size

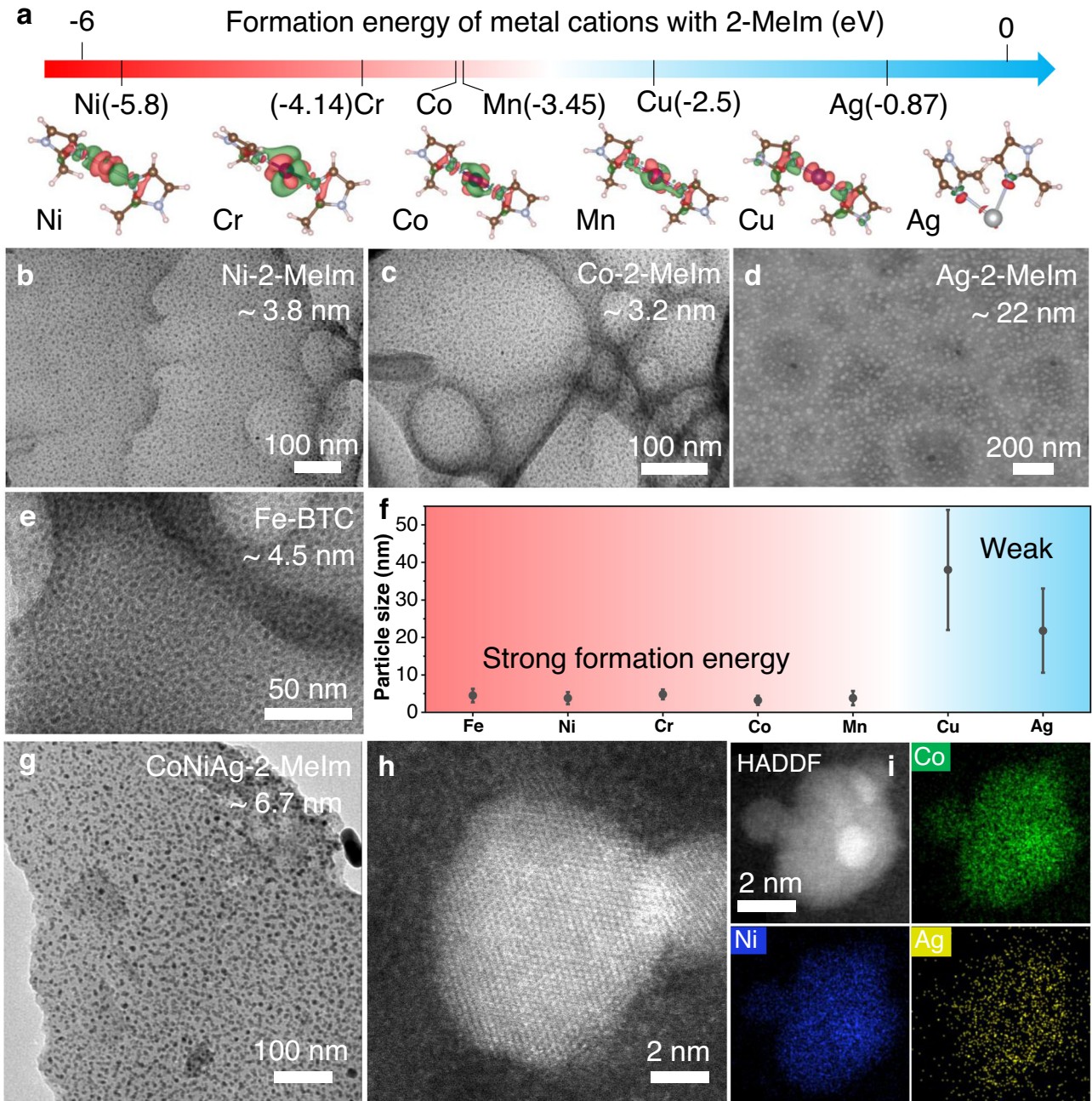

**Fig. 5 | General synthesis of single and multielement alloy NPs by coordinated CTS. a** The formation energy and charge distribution of metals Ni, Cr, Co, Mn, Cu, and Ag when coordinated with dimethylimidazole (2-MeIm). **b, c** TEM images of Ni-2-MeIm (3.8 nm) and Co-2-MeIm (3.2 nm), **d** SEM image of Ag-2-MeIm (22 nm), and **e** TEM image of Fe-benzenetricarboxylic acid (Fe-BTC) (4.5 nm) after CTS pyrolysis. **f** The particle size distribution of Fe, Co, Ni, Cu, Cr, Mn, and Ag by coordinated CTS (Error bars represent the standard deviation of 100 nanoparticles). **g–i** TEM, HRTEM, and EDS mapping images of CoNiAg synthesized by coordinated CTS pyrolysis.

and dispersion (Fig. 5b and Supplementary Fig. 45), indicating the generality of this method. However, note that $Fe^{3+}$ ions are easily hydrolyzed to form $Fe(OH)_3$ precipitates, and 2-MeIm (alkaline) promotes the formation of $Fe(OH)_3$ precipitates and therefore inferior dispersion (Supplementary Fig. 46). On the other hand, the formation energy of Ag and Cu with 2-MeIm is relatively weaker at −0.87 and −2.5 eV, respectively, which leads to larger particles due to insufficient coordination with 2-MeIm (Fig. 5d and Supplementary Fig. 43). We also demonstrated the 2D porous carbon films formed in the above systems (Supplementary Fig. 47).

Considering the huge flexibility in coordination chemistry, various other metal-ligand chemistries can be designed. The 1,3,5-

benzenetricarboxylic acid (BTC) ligand was used to coordinate with Cu and Fe, which did not show good results when coordinated with 2-MeIm. The formation energies of Cu and Fe with BTC are much stronger than those with 2-MeIm at −8.13 and −11.55 eV due to multiple coordination bonds formed between metal cations and oxygen groups (Supplementary Fig. 48 and Table 8). Such a strong coordination effect can restrict particle aggregation and achieve a uniform dispersion of NPs. Meanwhile, BTC (acid) can also restrain $Fe^{3+}$ hydrolysis and avoid precipitate formation. As shown in Fig. 5e and Supplementary Fig. 14, Cu-BTC and Fe-BTC displayed a uniform nanoparticle size and dispersion, despite the size of Cu-BTC after CTS pyrolysis being relatively large.

Similarly, the pyrolysis carbon films show a porous structure (Supplementary Fig. 49).

We summarized the relationship between particle size and coordination strength in Fig. 5f. Samples synthesized by coordinated CTS (Fe, Co, Ni, Cr, Mn) all exhibited a small size below 5 nm. However, when the coordination strength was weak (e.g., Cu and Ag), the particles were larger and had a wider distribution due to weak coordination. This highlights the importance of coordination bonds in confining and stabilizing metal clusters during pyrolysis to ensure uniform dispersion and anchoring. We also extended this synthesis to multielement alloy CoNiAg NPs (Fig. 5g), which had a similar ultrasmall size and uniform dispersion despite containing three different elements. In particular, Ag with weak coordination was effectively confined to small NPs due to Co and Ni incorporation. The HRTEM image of CoNiAg showed an alloyed solid solution structure (Fig. 5h), while the EDS mapping demonstrated uniform elements in the CoNiAg (Fig. 5i). Both DFT calculation and experimental results confirm that coordination bonds and their formation energy are critical for synthesizing high-density and ultrasmall NPs. This work can serve as a guideline for coordination chemistry design and catalyst synthesis using coordinated CTS pyrolysis.

In summary, we report a general and efficient method for synthesizing high-density and ultrasmall NPs dispersed and stabilized on 2D porous carbon. Our approach combines coordination chemistry design with rapid shock pyrolysis we call coordinated CTS process. This method is simple and fast, requiring neither lengthy preparation of MOFs or long pyrolysis in the furnace. Instead, it directly utilizes the metal-ligand precursor for rapid CTS pyrolysis within less than one second, during which the precursor self-assembled into a coordinated complex and then transformed into ultrasmall NPs stabilized on 2D porous carbon. Considering the great flexibility in coordination chemistry design, our method enables the general synthesis of various single and multielement NPs with great simplicity and versatility. The coordination design combined with rapid pyrolysis therefore affords a simple and universal method to synthesize high-density and ultrasmall NPs uniformly exposed on 2D porous carbon for a wide range of energy and environmental applications.

## Methods
### Materials and chemicals
Iron nitrate nine hydrates ($Fe(NO_3)_3 \cdot 9H_2O$, ACS Reagent Grade), cobalt nitrate hexahydrate ($Co(NO_3)_2 \cdot 6H_2O$, ACS Reagent Grade), nickel nitrate hexahydrate ($Ni(NO_3)_2 \cdot 6H_2O$, ACS Reagent Grade), copper nitrate trihydrate ($Cu(NO_3)_2 \cdot 3H_2O$, ACS Reagent Grade), chromium nitrate nonahydrate ($Cr(NO_3)_3 \cdot 9H_2O$, ACS Reagent Grade), manganese nitrate tetrahydrate ($Mn(NO_3)_2 \cdot 4H_2O$, ACS Reagent Grade), lanthanum nitrate hexahydrate ($La(NO_3)_2 \cdot 6H_2O$, ACS Reagent Grade), potassium hydroxide (KOH, ACS Reagent grade), ethanol ($CH_3CH_2OH$, ACS Reagent Grade), and dimethylimidazole ($C_5H_8N_2$, ACS Reagent Grade) were purchased from Sigma–Aldrich. Carbon paper was purchased from Toray Industry. The deionized water was purified on a MilliQ device from Millipore.

### Fabrication of high-density and ultrasmall nanoparticles
First, 8 mL 0.8 M 2-MeIm ethanol solution into an equal volume of 0.2 M $Co(NO_3)_2$ ethanol solution, which obtains the metal-ligand precursor solution. Then, a certain amount of metal-ligand precursor solution was uniformly dropped on the CP and dried at 60 °C in an oven overnight. The CP loaded with the metal-ligand precursor was fixed on the carbon plate. A large current pulse (10 A, 100 ms) was used for carbothermal shock in an Ar atmosphere. The Co-CTS sample was obtained after carbothermal shock. $Fe(NO_3)_3$, $Cu(NO_3)_2$, $Mn(NO_3)_2$, $Ni(NO_3)_2$, $AgNO_3$, $La(NO_3)_2$, $Cr(NO_3)_3$, and $Mg(NO_3)_2$ are also used as single or multicomponent metal salts. The whole

preparation process is similar to the above synthesis process for the Co-CTS sample. The carbon paper slice (length × width × thickness = 3.0 cm × 0.5 cm × 0.03 mm) is used as the heater. The current output mode is adopted and the maximum output voltage is 30 V. The fast heating rate of ~$10^4$ °C/s is achieved by an instant pulsing current to 10 A in 100 ms (~100 A/s), and other slower heating rates (100 and 1000 °C/s) can be achieved by programming the current stepwise at rates of 1 A/s and 10 A/s from 0 to 10 A. Finally, a heating rate of 0.1 °C/s is obtained by conventional furnace treatment (6 °C/min).

### Structural characterization
SEM was performed at 10 kV on a TESCAN MIRA4 with EDS analysis at 15 kV. XPS was carried out on a Thermo Scientific K-Alpha. XRD was performed on a Bruker C2 Discover X-ray powder diffractometer. The temperature of CP with the MOC precursor during the high-temperature shock process was recorded on a Fluke Process Instrument E1RH-R59-V-0-0. TEM images were obtained using an aberration-corrected FEI Titan Themis G2 300 microscope. Thermogravimetric analysis (TGA) was performed by Diamond TG/DTA (PerkinElmer Instruments) at a heating rate of 20 °C/min in Ar gas. Fourier transform infrared (FTIR) spectra were collected using a Nicolet iS50R (Thermo Scientific). The Co $K$-edge XANES data were recorded in transmission mode and analyzed using the IFEFFIT program. Co foil, CoO, and $Co_3O_4$ were used as references.

### Electrochemical characterization
The electrochemical performance was tested in a three-electrode setup by a CS3104 electrochemical workstation. The graphite rod, Ag/AgCl electrode with saturated potassium chloride solution, and as-prepared catalysts were used as the counter, reference, and working electrodes, respectively. The electrolyte was 0.1 M and 1 M KOH solution for the ORR and OER tests, respectively.

For the ORR test, the catalysts (prepared on carbon paper heater) were sonicated in ethanol to form a uniform solution and then dried. The catalyst ink was prepared by dispersing 5 mg catalyst powder in a mixed solution of 750 µL isopropanol, 200 µL water, and 50 µL 5 wt% Nafion 117 solution, followed by sonicated for 30 min to get a homogeneous suspension. Then 10 µL of the as-obtained catalyst ink was loaded onto the surface of glassy carbon (GC) electrode with ~5 mm in diameter. The catalyst was first activated via the cyclic voltammetry (CV) method from 0.3 to 1.1 V at scan rates of 50 mV/s for 100 cycles. LSV curves were investigated from 0.4 V to 1.1 V at scan rates of 10 mV/s. Chronoamperometry and accelerated cyclic voltammetry tests were used to test the stability of the catalyst. The electrolyte resistance is 20.3 Ω measured by EIS.

For the OER test, the electrode test area is 0.5 × 0.5 $cm^2$. the activation process was carried out from 0 to 0.6 V at scan rates of 50 mV/s for 100 cycles. LSV curves were recorded from 0 to 1 V at a scan rate of 5 mV/s. The electrochemically active surface area (ECSA) of these samples was determined by measuring the double-layer capacitance ($C_{dl}$) in a nonfaradaic region. Typically, CV scans were carried out with scan rates ranging from 20 to 100 mV/s in the potential range from 0 to 0.1 V versus RHE. By plotting the capacitive current at 0.05 V versus RHE against the scan rates, the $C_{dl}$ was obtained as half of the corresponding slope, and the ECSA was derived from the equation ECSA = $C_{dl}/C_s$, where $C_s$ is the specific capacitance. The $C_s$ for a flat surface was in the range of 20–60 µF/$cm^2$, and a value of 40 µF/$cm^2$ was adopted here to roughly calculate the ECSA. The stability was measured at a current density of 10 mA/$cm^2$ for 12 h. The electrolyte resistance is 3.2 Ω measured by EIS.

All potentials were compensated by 90 % iR to correct the ohmic resistance of the solution, $E_{corrected} = E - 0.9 \times iR$. The corrected potential values are −7.2 mV at 10 mA/$cm^2$ in OER test and 8.96 mV at

$E_{1/2}$ in ORR test, respectively. The tested potentials versus Ag/AgCl were converted to reversible hydrogen electrodes (RHE) based on the following equation:

$$E_{RHE} = E_{Ag/AgCl} + 0.059 \times pH + 0.197\,V \tag{1}$$

The pH values of electrolytes value were tested by a pH meter (FE28, Mettler Toledo, Switzerland). For 1 M KOH electrolyte, its pH value is $13.80 \pm 0.15$; For 0.1 M KOH electrolyte, its pH value is $12.83 \pm 0.12$.

## Zinc-air battery test

The batteries performances were tested with homemade zinc-air battery using CS3104 electrochemical workstation. The catalyst ink was drop-casted onto carbon paper, then dried at room temperature, the catalyst loading was 1 mg/cm². A Zinc foil was used as an anode. The zinc-air battery was then assembled by filling the electrolyte (6 M KOH and 0.2 M $Zn(Ac)_2$) between the anode and the air-cathode. The cycling test was performed at 10 mA/cm² over 30 min discharge followed by 30 min charge per cycle, using a LAND battery testing workstation at ambient air condition.

## DFT calculation

First-principles calculations for structural relaxation and formation energy were performed using density-functional theory (DFT) as implemented in the ORCA quantum chemistry program (version 5.3)[72,75,76]. We employed the Becke-Perdew86 (BP86) density function and expanded the molecular wave functions through def2-type split valence polarization basis sets (def2-SVP)[73,74]. The atom-pairwise dispersion corrections (D3) were considered to account for weak intermolecular interactions[71]. To achieve high accuracy for both the self-consistent-field (SCF) iterations and geometry optimization, we set tight SCF convergence criteria (single-point energy change within $10^{-7}$ eV) and perform vibrational frequency calculations to verify that the molecular geometries arrive at local minima of potential energy surfaces.

Fixing atomic positions according to optimized molecular geometries, we calculated the ground state electronic energies and charge densities of the metal ion, the ligands and the metal-ligand complex. The formation energies $\Delta E$ and the charge density differences $\Delta\rho(\boldsymbol{r})$ are then obtained by

$$\Delta E = E_{\text{metal–ligands}} - \left( E_{\text{metal}} + E_{\text{ligands}} \right) \tag{2}$$

and

$$\Delta\rho(\boldsymbol{r}) = \rho_{\text{metal–ligands}} - \left( \rho_{\text{metal}} + \rho_{\text{ligands}} \right) \tag{3}$$

respectively. For different kinds of metal atoms with the same ligands, the relative strengths of the coordination effect can be quantitatively described by the formation energy. As shown in Fig. 4a, e, charge density differences spatially exhibit electron gain and loss from metals to ligands. The red zone represents electron gain, while the green zone denotes electron loss. The alternating distributions of the electron gain zone and loss zone within the metal ion and N (O) atoms imply charge transfer induced by the coordination effect. Here, the isosurface levels for charge density are consistently set to 0.005e/Bohr³. The formation energy and integral of the charge difference densities straightforwardly indicate the strengths of the coordination bonds.

## Data availability

Relevant data supporting the key findings of this study are available within the article and the Supplementary Information file. All raw data generated during the current study are available from the corresponding authors upon request.

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

## Acknowledgements

This work is supported by the National Key R&D Program of China (2021YFA1202300 (Y.Y.)), the National Natural Science Foundation of China (52101255 (Y.Y.), 22172003 (J.Z.), 22271110 (X.C.)), the Knowledge Innovation Program of Wuhan - Shuguang Project (0216110173 (Y.Y)), the China Postdoctoral Science Foundation (2022M722646 (Y.-C.H.)), and the Fundamental Research Funds for the Central Universities, HUST: 2021GCRC046 (Y.Y). The authors are thankful for the test support from the Analytical and Testing Center of Huazhong University of Science & Technology, the State Key Laboratory of Materials Processing and Die & Mould Technology, and the Electron Microscopy Laboratory at Peking University for the use of the aberration-corrected electron microscope.

## Author contributions

Z.-Q.T., Y.Y., and B.X. conceived and designed the project. W.S. synthesized the materials. W.S. and H.L. carried out the electrochemical testing. W.S., Z. Li, Z. Liang, J.Z., Y.-C.H., X.C., B.X., Y.Y., and Z.-Q.T. contributed to the characterization and related discussion. W.S. and Z. Liang performed the in situ GIWAX measurements. Z.G. and H.W. contributed to the theoretical calculations. H.N., B.S., and B.X. contributed to the EXAFS analysis. Z. Li and J.Z. contributed to the STEM measurements. W.S., Y.-C.H., X.C., B.X., Y.Y., and Z.-Q.T. co-wrote the paper. All the authors discussed the results.

## Competing interests

The authors declare no competing interests.

## Additional information

¹State Key Laboratory of Materials Processing and Die & Mould Technology, School of Materials Science and Engineering, Huazhong University of Science and Technology, 430074 Wuhan, China. ²Beijing National Laboratory for Molecular Sciences, College of Chemistry and Molecular Engineering, Peking University, 100091 Beijing, China. ³ZJU-Hangzhou Global Scientific and Technological Innovation Center, School of Micro-Nano Electronics, Zhejiang University, 311200 Hangzhou, China. ⁴State Key Laboratory of Physical Chemistry of Solid Surfaces, College of Chemistry and Chemical Engineering, Innovation Laboratory for Sciences and Technologies of Energy Materials of Fujian Province (IKKEM), Xiamen University, 361005 Xiamen, China. ⁵Key Laboratory of Material Chemistry for Energy Conversion and Storage (Ministry of Education), School of Chemistry and Chemical Engineering, Huazhong University of Science and Technology, 430074 Wuhan, China. ✉e-mail: byxia@hust.edu.cn; yaoyg@hust.edu.cn; zqtian@xmu.edu.cn

