## [Peer review file · Nature Communications]

REVIEWER COMMENTS

Reviewer #1 (Remarks to the Author):

In this manuscript, the authors synthesized the high-density and ultrasmall nanoparticles uniformly dispersed on two-dimensional porous carbon via directly coordinated carbothermal shock (coordinated CTS, ~ 100 ms) of metal-ligand precursors (e.g., Co^{2+} and dimethylimidazole $\text{C}_5\text{H}_8\text{N}_2$). The as-prepared Co-CTS sample exhibited outstanding electrochemical performance toward ORR and ER. Furthermore, the authors studied the effect of organic ligands and coordination chemistry during the rapid CTS process. Although the characterization and mechanism study for the formation of the Co-CTS sample are well-organized, some detailed discussion on the enhanced catalytic performance of the Co-CTS sample is needed. This manuscript is acceptable for publication in Nature Communications after the following points are addressed.

1. The authors explained that the Co NPs are surrounded by the N atoms, suggesting the N-stabilized Co NPs after the CTS process. To support this encapsulated morphology of Co and N atoms in Co-CTS, it would be better to show the HRTEM image and analyze the d-spacing for the Co and N species.
2. Gases such as NO_x , CO_x , and H_2O play the role of pore-forming agents for the formation of Co-CTS. Are there different effects stemming from these gases, such as pore size and uniformity of hole size?
3. Is there any electron transfer effect between Co and N atoms in the Co-CTS sample? Did the electron transfer effect enhance the catalytic performance of Co-CTS toward ORR or OER? The authors should add the Co XPS spectra of Co-CTS as well as the N XPS spectra in Figures 2f and S15 to present the oxidation state of Co species in Co-CTS. Furthermore, to verify the exact tendency for the chemical state of the catalyst surface, I suggest that the authors provide the table in Supporting information, which summarizes the relative peak areas (%) for each Co and N species and the parameters used to fit the XPS spectra, for example, binding energy and FWHM.

Moreover, to support the electron transfer effect between Co and N atoms, it would be better to add another X-ray analysis, namely, X-ray absorption spectroscopy (XAS) analysis, which can confirm the charge change of the atoms in nanoparticles.

4. The authors demonstrated that the Co-CTS ($326 \text{ m}^2 \text{ g}^{-1}$) has a higher Brunauer-Emmett-Teller (BET) specific surface area than the Co-furnace sample ($190 \text{ m}^2 \text{ g}^{-1}$) by the nitrogen adsorption and desorption method. These pores are critical for nanoparticle exposure and electrolyte penetration to largely improve performance in electrochemical reactions. Is there any pore size effect on electrocatalytic performance?
5. In Figure 3e, the Co-CTS with 2 Melm catalyst (280 mV to drive a current density of 10 mA cm^{-2}) exhibited higher OER activity than Co-CTS without 2 Melm (330 mV to drive a current density of 10 mA cm^{-2}). However, at the potential region from 1.5 V to 1.7 V , there is no significant difference in the OER activity of the two catalysts (with and without 2 Melm) because the reaction kinetic of Co-CTS with 2

Melm catalyst slows down as the potential increases. Therefore, the authors should measure the reaction kinetics during the OER operation of prepared catalysts by calculating the Tafel slope.

6. Furthermore, stability is a critical factor to demonstrate the novelty of the prepared catalysts toward ORR and OER. The authors should conduct the stability test (chronopotentiometry or chronoamperometry test) or durability cycle (ADT test) to prove the stability of the Co-CTS catalyst. Moreover, the structural characterizations (TEM, XRD, XPS, etc.) for the catalyst after the stability test or durability cycle are needed to understand the change of structural property and chemical states of the catalyst surface and prove the stability of the prepared catalyst. This analysis might help explain why the ORR and OER performance was enhanced by Co-CTS with 2 Melm catalyst. Therefore, the authors should provide durability or stability measurement results of prepared catalysts and additional characterizations after the durability or stability test with a detailed discussion about the change of crystal structure and oxidation structure after long-term operation.

Reviewer #2 (Remarks to the Author):

In this work, Shi et al. reported the synthesis of high-density of ultrasmall nanoparticles by carbothermic shock. Metal salts and $C_2H_8N_2$ were mixed and loaded onto carbon black. The rapid Joule heating pyrolyzed the precursors and nanoparticles were formed. The main conclusion claimed in this work is the high density and ultrasmall size of the obtained nanoparticles. They authors demonstrated the application as bifunctional oxygen electrocatalysts. The main concern from the reviewer is that the carbothermic shock synthesis of similar metal nanoparticles is very well documented. The novelty of this work might not be warranted to publish in a Nature series journal.

1. Tremendous works were published recently to use the carbothermic shock synthesis of single-metal nanoparticles, bimetallic nanoparticles, high-entropy nanoparticles, etc. The main conclusion of this work is the high density and small size. However, with regard to high density, Song et al. reported the generation of high-density nanoparticles in the carbothermal shock method (Sci. Adv. 7, eabk2984 (2021)), with a similar strategy to enhance the metal-carbon substrate interaction. They achieved a coverage of 85%, which is higher than this work is showing. With regard to ultrasmall size, Lacey et al. reported the multimetallic nanoparticle size of 2-3 nm by the carbothermic shock method (Nano Lett. 2019, 19, 8, 5149–5158), which is smaller than the values the authors reported here (3.2nm – 22nm, Figure 5). The same strategy of rapid Joule heating pyrolysis of MOF for nanoparticles synthesis has already reported by the authors (Nano Energy, 2022, doi: 10.1016/j.nanoen.2022.107125). In that work, they reported sub-3 nm Co nanoparticle with high loading up to 41%, where the particles size is, again, smaller than the values reported here. Hence, I cannot see substantial advance of this work over previous publications.

2. The authors used the Co nanoparticles in ORR and OER. Co nanoparticles as electrocatalysts has been widely reported. The authors should compare their materials performances with literature values and present that for a comparison for the readers to see. Does the high density and small size enhance the performance?
3. The measurement conditions of TGA (Fig. 4a, Fig. S23) should be provided. Was it measured in inert gas or air? What is the heating rate? The y-axis values are missing for these TGA plots.
4. The authors show the synthesis at different heating rates of 0.1, 10, 100, 10^4 °C/s. How is the heating rate controlled by the CTS process?
5. What is the chemical state of Co? Is it Co(0) or with some oxide composition? XPS or other analysis should be provided. How does the difference affect the electrocatalytic performance?

Reviewer #3 (Remarks to the Author):

In this manuscript, the authors report a new synthesis method of high-density nanoparticles (NPs) on 2D porous carbon based on combination of coordination chemistry design and carbo-thermal shock (CTS). The prepared Co NPs in this manuscript exhibits ultra-small NPs with high density, and shows high OER, ORR and Zn-air performances. The concept of coordination chemistry for generation of high density NPs using CTS is proved by matching other metal (e.g. Cu) and ligand (BTC). Overall, this manuscript possesses novelty to some extent especially in CTS method, and it is well organized with comprehensive characterizations. Therefore, this manuscript can be accepted after addressing the following issues:

1. To arouse a broad interest from readership in this field, some important and closely related literature about recent progress of generation of high density NPs using CTS should be cited in the revised manuscript (e.g. Sci. Adv. 2021, 7, eabk2984; ACS Appl. Mater. Interfaces 2019, 11, 29773; Nat. Commun. 2020 11, 6373). Compared to previous studies, the originality and advancement of this manuscript should be further elaborated in the Introduction.
2. The authors should represent morphology of Co NPs (e.g. SEM and/or TEM images) after electrochemical stability test.
3. Although the main point of this paper is introducing the formation of high-density NPs by combination of CTS and coordination chemistry, if there are some mechanistic studies of the ORR and OER, that would be very helpful and useful.

Point-by-point responses to reviewers' remarks

(Black italic: Reviewer's comments; Blue type: Our response; Red type: Our revised)

Note: To clearly mark our samples for easy understanding, we have listed their abbreviations as shown below. Specifically, we added N (coordination element) into samples prepared from metal-ligand precursors to highlight the importance of N coordination.

Sample names	Sample detail information
Co-N-CTS	CTS pyrolysis of Co(NO ₃) ₂ and 2-MeIm precursor
Co-N-CTS-X (eg., Co-N-CTS-1000)	CTS pyrolysis of Co(NO ₃) ₂ and 2-MeIm precursor at X °C
Co-CTS	CTS pyrolysis of Co(NO ₃) ₂ without 2-MeIm precursor
Co-N-furnace	Furnace pyrolysis of ZIF-67 (Co and 2-MeIm ligand)

Reviewer#1:

In this manuscript, the authors synthesized the high-density and ultrasmall nanoparticles uniformly dispersed on two-dimensional porous carbon via directly coordinated carbothermal shock (coordinated CTS, ~100 ms) of metal-ligand precursors (e.g., Co^{2+} and dimethylimidazole $\text{C}_5\text{H}_8\text{N}_2$). The as-prepared Co-CTS sample exhibited outstanding electrochemical performance toward ORR and OER. Furthermore, the authors studied the effect of organic ligands and coordination chemistry during the rapid CTS process. Although the characterization and mechanism study for the formation of the Co-CTS sample are well-organized, some detailed discussion on the enhanced catalytic performance of the Co-CTS sample is needed. This manuscript is acceptable for publication in Nature Communications after the following points are addressed.

Response: We sincerely appreciate the reviewer for the valuable comments, which have made a tremendous improvement to our manuscript. Specifically, we have added a detailed discussion on the enhanced catalytic performance of the Co-N-CTS sample, including new structural characterization data (XPS and XAS (XANES and EXAFS)), electrochemical analysis of ECSA-normalized catalytic activity and the Tafel slope, and a separate paragraph to discuss the structure-property relationship. We hope that these revisions address the concerns, and we truly thank you for your time and efforts.

1. The authors explained that the Co NPs are surrounded by the N atoms, suggesting the N-stabilized Co NPs after the CTS process. To support this encapsulated morphology of Co and N atoms in Co-CTS, it would be better to show the HRTEM image and analyze the d-spacing for the Co and N species.

Response: We are grateful for the reviewer's valuable comment about the Co-N structure in our Co-N-CTS sample. First, the statement "Co NPs are surrounded by the N atoms" is from the description of our EDS data (**Figure 2e**, also shown below), where we found Co and N elements are closely correlated spatially. To clearly illustrate the detailed structure in our Co-N-CTS sample, following the suggestion of the reviewer, we conducted additional experiments to analyze the structure, as detailed below.

1. HAADF-STEM and EDS mapping: **Figure R1** shows the high-angle annular dark-field scanning TEM (HAADF-STEM) and corresponding EDS mapping images of the Co-N-CTS sample. Co is mainly distributed in the nanoparticle; N is distributed in not only the Co nanoparticle but also the carbon region, indicating the formation of N-doped carbon from the pyrolysis of the 2-MeIm ligand ($\text{C}_5\text{H}_8\text{N}_2$), while C is distributed in the whole scanning area. Particularly, from the overlap of Co-N mapping, it is clear that Co and N are spatially correlated, indicating that Co nanoparticles are dispersed and stabilized by the N-doped carbon. We also added a higher resolution EDS mapping, showing that the Co nanoparticles are distributed on the N-doped carbon (**Figure R2**).

Figure R1. HAADF and EDS mapping images of Co-N-CTS.

Figure R2. High-magnification HAADF-STEM and EDS images of Co-N-CTS.

2. **HRTEM:** Following the reviewer's suggestion, we also performed HRTEM on the Co-N-CTS sample. As shown in **Figure R3**, uniform and small nanoparticles were densely distributed. The lattice fringe spacing is 0.21 nm, which can be assigned to the metallic Co (111) crystal plane.

Figure R3. High-resolution TEM image of Co-N-CTS.

3. **XPS**: To further analyze the Co-N correlation, X-ray photoelectron spectroscopy (XPS) was used for the Co-N-CTS sample to characterize their valence states and bonds. We especially focused on the N peak deconvolution (**Figure R4**), which shows not only pyridinic, pyrrolic, graphitic, and oxidized N but also Co-N peaks (peak deconvolution references: *Adv. Funct. Mater.* 2019, 29, 1904481, *Adv. Funct. Mater.* 2021, 31, 2104735). The relative peak area and key fitting parameters are also summarized in **Table R1**. The Co-N bond and strong coordination indicate strong metal support interaction (SMSI), which could result in an encapsulation overlayer (N-doped carbon) to stabilize the Co NPs by preventing the agglomeration of Co NPs (*Applied Catalysis B: Environmental* 281, 2021, 119514; *Adv. Sci.* 2021, 8, 2101438). In addition, the existence of strong Co-N coordination is further proven by XAS analysis (please refer to **Comment #3**).

Figure R4. High-resolution XPS deconvolution of the N element of Co-N-CTS.

Table R1. The peak areas (%), binding energy, and FWHM for N species in Co-N-CTS.

	pyridinic N	Co-N	pyrrolic N	graphitic N	oxidized N
Binding energy (eV)	398.22	399.02	400.25	401.1	402.16
FWHM	1.52	1.85	1.3	1.23	0.92
Peak areas (%)	33.5	27.2	14.1	19.8	5.4

In summary, the Co-N-CTS sample was synthesized by direct CTS pyrolysis of a coordinated Co-2-MeIm ($C_5H_8N_2$) precursor. Owing to the formation of the Co-N coordination bond and rapid pyrolysis, ultrasmall Co nanoparticles are formed and uniformly distributed on N-doped carbon. Due to the strong metal-substrate interaction, the Co nanoparticles are stabilized by an encapsulation overlayer (N-doped carbon), which is proven by HAADF-STEM, EDS mapping (Co-N spatial correlation), HRTEM (fuzzy shell), and XPS (and later XAS) spectra data showing Co-N bonds and coordination.

Modification in the revised manuscript:

Following the reviewer’s suggestion, to help the reader’s understanding of the Co-N structure in our Co-N-CTS sample, we added more detailed discussions in our revised manuscript.

Page 7, lines 13-15.

The high-resolution TEM confirms the high-density distribution of Co NPs, showing a lattice fringe spacing of 0.21 nm, which can be assigned to the metallic Co (111) crystal plane

(Figure S20).

Page 7, lines 16-23.

Moreover, high-angle annular dark-field scanning TEM (HAADF-STEM) and corresponding EDS mapping of the Co-N-CTS sample are also investigated. As shown in Figure 2e and S21, Co is mainly distributed in the nanoparticle; N is distributed in not only the Co nanoparticle but also the carbon region, indicating the formation of N-doped carbon from the pyrolysis of the 2-MeIm ligand (C₅H₈N₂). Particularly, from the overlap of Co-N mapping, it is clear that Co and N are spatially correlated, indicating that the Co NPs are dispersed and stabilized by the N-doped carbon, originating from the strong metal-substrate (Co-N) interaction, as will be discussed later.

Modification in revised Supporting Information:

In addition, high magnification HAADF-STEM, EDS, HRTEM image, and the fitting parameters of XPS are also added as Figure S20, S21, and Table S2 in Supporting Information as suggested by the reviewer.

2. Gases such as NO_x, CO_x, and H₂O play the role of pore-forming agents for the formation of Co-CTS. Are there different effects stemming from these gases, such as pore size and uniformity of hole size?

Response: We appreciate the reviewer's comment. We agree with the reviewer that these gases generated during the CTS process play a key role in pore formation in our Co-N-CTS samples. Following the reviewer's suggestion, to confirm whether different gases can induce different effects on pore size and uniformity, we comparably studied two samples, precursor Co with dimethylimidazole ligand (Co-2-MeIm) and precursor Cu with benzene tricarboxylic acid ligand (Cu-BTC) (Figure R5), by CTS pyrolysis, where ligands with different compositions will lead to different gas generation.

Figure R5. Schematic diagram of 2-MeIm and BTC ligands.

1. **Effect of gas species.** After CTS pyrolysis, we characterized the pore formation by SEM and BET. As shown in Figure R6a and R6d, the samples derived from Co-2-MeIm and Cu-BTC all display a pore-forming microstructure, indicating that coordinated CTS is effective and general in inducing an open porous structure, while the difference in gas species does not obviously affect the formation of pores. As shown in Figure R6, the BET of Co-2-MeIm and Cu-BTC by CTS all showed a larger surface area of 326 and 221.9 m²/g than that by furnace annealing. In addition, the pore sizes of samples by CTS are in a much broader size range (i.e., hierarchical porosity) compared with the corresponding samples by furnace annealing with mostly micropores. These results indicate that the fast heating rate (or fast gas generation) by CTS pyrolysis leads to explosive carbonization and promotes the formation of hierarchical pores, while the differences in gases may play a secondary role in the pore formation process.

Figure R6. SEM image, N_2 adsorption-desorption isotherms, and corresponding pore size distribution of Co-N-CTS (a-c) and Cu-O-CTS (d-f).

2. **Effect of heating rate:** In our original manuscript, we studied the heating rate effect on pore formation in the Co-N-CTS sample, which is believed to be critical to creating open porous structures and thin carbon films. As shown in **Figure R7**, a compact and dense structure was obtained with slow heating ($0.1 - 100 \text{ }^\circ\text{C/s}$), while increasingly porous and open structures were obtained with an increased heating rate ($1000 - 10^4 \text{ }^\circ\text{C/s}$). However, only at a sufficiently high heating rate, e.g., $10^4 \text{ }^\circ\text{C/s}$, do we obtain a porous structure that is ideal for nanoparticle exposure and electrolyte penetration, thus enhancing the catalytic activity and particularly the kinetics at high overpotentials.

Figure R7. SEM images of MOC at different heating rates of 0.1 , 100 , 1000 , and $10^4 \text{ }^\circ\text{C/s}$.

We also used the electrochemical surface area (ECSA) to evaluate the exposed surface area and active sites in the synthesized Co-N-CTS samples. As shown in **Figure R8a** and **R8b**, the ECSA increases as the heating rate increases, confirming that the faster heating rate benefits open and porous structure formation and more active site exposure. The heating rate effect on the pore formation of Cu-BTC (**Figure R8c** and **R8d**) shows a similar result, confirming the critical rate dependence of pore formation in the coordinated CTS process.

Figure R8. The C_{dl} values and ECSA of (a and b) Co-2-MeIm and (c and d) Cu-BTC by furnace and CTS treatment at different heating rates.

3. Effect of heating temperature: The heating temperature effect on the porous structure and particle size is also investigated. As shown in **Figure R9**, open and porous structures were obtained in the CTS process from ~ 600 to 1200 °C owing to the fast heating rate (10^4 °C/s), while the particle size increased with increasing temperature, further suggesting that the heating rate is critical to creating open porous structures, and the heating temperature can tune the particle sizes.

Figure R9. (a-d) SEM and (e-f) TEM images of Co-N-CTS samples pyrolyzed at different temperatures (600, 800, 1000, and 1200 °C).

In summary, different gases (e.g., from different ligands) generated from coordinated CTS can result in differences in surface area and pore distribution. However, it seems that the heating rate in coordinated CTS pyrolysis is much more important in determining the surface area and pore size distribution. Usually, a higher heating rate leads to a larger surface area and increased exposure of active sites, which validates the advantages of coordinated CTS pyrolysis.

Modification in the revised manuscript:

Following the reviewer's valuable suggestion, to help the reader's understanding of pore formation, we added more detailed discussions in our revised manuscript.

Page 6, lines 20-23; page 7, line 1-7.

We therefore studied the heating rate effect (from ~ 0.1 to 10^4 °C/s), which is believed to be critical to creating hierarchical porous structures and thin carbon supports, as revealed by SEM, BET, and electrochemically active surface area (ECSA, **Figure S8-S11**). A compact and dense structure was obtained with slow heating (0.1 - 100 °C/s), while increasingly porous and open structures were obtained with an increased heating rate. However, only at a sufficiently high heating rate, e.g., 10^4 °C/s, do we obtain an open porous structure that is ideal for nanoparticle exposure, electrolyte penetration, and mass/gas transportation (**Figure S12**). The critical heating rate dependence of pore formation is further confirmed in Cu with a benzene tricarboxylic acid ligand (Cu-BTC) precursor (**Figure S13-S16**). Meanwhile, the pyrolysis temperature will also induce changes in the porous structure and nanoparticle size (**Figure S17**), crystallinity (**Figure S18a**), and graphitization (**Figure S18b**).

Modification in revised Supporting Information:

In addition, BET and ECSA are also added to **Figures S14-S16** in the Supporting Information as suggested by the reviewer.

3. Is there any electron transfer effect between Co and N atoms in the Co-CTS sample? Did the electron transfer effect enhance the catalytic performance of Co-CTS toward ORR or OER? The authors should add the Co XPS spectra of Co-CTS as well as the N XPS spectra in Figures 2f and S15 to present the oxidation state of Co species in Co-CTS. Furthermore, to verify the exact tendency for the chemical state of the catalyst surface, I suggest that the authors provide the table in Supporting information, which summarizes the relative peak areas (%) for each Co and N species and the parameters used to fit the XPS spectra, for example, binding energy and FWHM.

Moreover, to support the electron transfer effect between Co and N atoms, it would be better to add another X-ray analysis, namely, X-ray absorption spectroscopy (XAS) analysis, which can confirm the charge change of the atoms in nanoparticles.

Response: We appreciate the reviewer's insightful comment. Generally, electrons can transfer from one element of weak electronegativity to another element of strong electronegativity, which can accordingly modulate their electronic configuration and associated catalytic properties. To prove and characterize the electron transfer, following the reviewer's suggestion, we performed a detailed analysis of Co and N in our Co-N-CTS samples.

I. XPS: Accordingly, we obtained the Co XPS spectra of Co-N-CTS and Co-CTS. As shown in **Figure R10**, a strong Co^0 signal was detected at 778.5 eV, and weak Co^{2+} and Co^{3+} signals were also observed, indicating surface oxidation of Co NPs (*Angew. Chemie*, 2019, 131, 2648–2652, *Nano Energy* 2018, 52, 485–493). Additionally, the peak position of metallic Co shifts to a higher binding energy region in Co-N-CTS than in Co-CTS, indicating that partial electron transfer from Co to N and N coordination effectively modulate the electronic structure of Co (*Adv. Mater.* 2017, 29, 1700874). Additionally, the relative peak area and key fitting parameters

are also summarized in **Tables R2** and **R3**.

Figure R10. The Co 2p XPS spectra of Co-N-CTS and Co-CTS.

Table R2. The peak areas (%), binding energy, and FWHM for Co species in Co-N-CTS.

	Co ⁰ 2p3/2	Co ³⁺ 2p3/2	Co ²⁺ 2p3/2	Satellite peaks	Co ⁰ 2p3/2	Co ³⁺ 2p3/2	Co ²⁺ 2p3/2
Binding energy (eV)	778.41	779.83	781.44	783.76	793.42	795.08	796.36
FWHM	1.33	1.99	2.37	3.5	1.86	1.63	1.63
Peak areas (%)	34.5	20.0	11.4	12.1	12.8	5.5	2.8

Table R3. The peak areas (%), binding energy, and FWHM for Co species in Co-CTS.

	Co ⁰ 2p3/2	Co ³⁺ 2p3/2	Co ²⁺ 2p3/2	Satellite peaks	Co ⁰ 2p3/2	Co ³⁺ 2p3/2	Co ²⁺ 2p3/2	Satellite peaks
Binding energy (eV)	778.22	779.39	781.48	784.28	793.36	795.19	796.86	903.19
FWHM	1.16	2.33	2.72	3.5	2.02	2.17	2.95	3.5
Peak areas (%)	22.7	22.7	16.0	8.4	12.5	4.5	8.2	5.0

2. XAS: To further investigate the electronic structure and coordination environment of Co species in Co-N-CTS, Co K-edge X-ray absorption near-edge structure (XANES) spectroscopies and extended X-ray absorption fine structure (EXAFS) analysis were also conducted (*Energy Environ. Sci.* 2018, 11, 2945–2953, *Nat. Commun.* 2018, 9, 2885). For comparison, Co foil, CoO, Co₃O₄, and Co-CTS were used as the reference samples. The Co K-edge in the XANES spectra (**Figure R11a**) shows that the absorption near edge spectrum for Co-N-CTS lay between those for Co foil and CoO, indicating that the partial electrons transfer from Co to N, and the valence state of Co species is between 0 and +2. However, the absorption near edge spectrum of Co-CTS lay near that for Co foil, indicating that the valence state of Co species is near 0 (metallic) and lacks electron transfer between Co and C substrate.

Additionally, the Fourier transformed EXAFS (FT-EXAFS) spectra of the Co K-edge for Co-N-CTS further indicate the coexistence of Co-N/O and metallic Co-Co scattering paths (**Figure R11b**). For Co-CTS, only a predominant peak at ≈ 2.16 Å was observed, which is

typically assigned to the Co nanoparticles. (*Angew. Chem.Int. Ed.* 2023, 62, e202214988, *Small*2022, 18, 2105487.)

Figure R11. (a) Normalized XANES spectra at the Co K-edge of Co-N-CTS, Co-CTS, and reference samples. The inset shows the enlarged marked area. (b) k^3 -weighted Fourier transform of EXAFS spectra at the Co K-edge of Co-N-CTS, Co-CTS, and reference samples.

3. Charge density distribution: We also theoretically calculated the charge density distribution of Co on carbon and Co on N-doped carbon. As shown in **Figure R12**, the Co on the carbon lost electrons ($0.215 e^-$), while after N was doped into carbon, the Co on N-doped carbon transferred/lost more electrons ($0.483 e^-$), indicating stronger electron transfer.

Figure R12. the difference in charge density between Co and the rest of the system. Yellow represents the electron accumulation area, and cyan is the electron depletion area. Isosurface level: $0.002 e^-/\text{Bohr}^3$.

4. Catalytic performances: From the above analysis, it is clear that the Co species in the Co-CTS and the Co-N-CTS samples have significantly different electronic structures and coordination environments. Therefore, we compared the catalytic performances of these two samples to illustrate the catalytic differences caused by the different Co species in the Co-CTS

and the Co-N-CTS samples (**Figure R13**). Obviously, Co-N-CTS exhibited better ORR and OER performances than the Co-CTS sample. Particularly for the ORR, Co-N-CTS has a much higher $E_{1/2}$, attributed to the strong interaction between Co NPs and N-doped carbon, and their electron transfer significantly regulates the electronic structure of Co to optimize the binding energies toward intrinsically enhanced catalytic activity (*Angew. Chemie Int. Ed.* 2019, 58, 12540-12544, *ACS Catal.* 2020, 10, 12376-12384).

Figure R13. Comparison of the (a) ORR and (b) OER performances of the Co-CTS and Co-N-CTS samples.

In summary, the Co 2p XPS spectrum, the XAS measurement, particularly the pre-edge XANES analysis, and the charge density distribution all point to partial electron transfer from Co to N in our Co-N-CTS samples, which is crucial in enhancing its intrinsic catalytic activity.

Modification in the revised manuscript:

Following the reviewer's valuable suggestion, to help the reader's understanding of the electron transfer between Co and N atoms in the Co-N-CTS sample, we updated the XAS data in **Figure 2** and added more detailed discussions in our revised manuscript.

Page 8, lines 21-23; page 9, line 1 and 7.

In addition, by detailed analysis of the Co XPS spectra (**Figure S28, Tables S3 and S4**), a strong Co^0 signal was detected at 778.5 eV, while weak Co^{2+} and Co^{3+} signals were also observed, suggesting surface oxidation of Co NPs^{48,49}. Additionally, the peak position of metallic Co 2p shifts to a higher binding energy (~ 0.3 eV) in our Co-N-CTS sample, indicating partial electron transfer from Co to N and effective modulation of the electronic structure in Co-N-CTS by the strong metal-substrate interaction between Co NPs and N-doped carbon⁵⁰. The metal-substrate interaction and electron transfer are further verified by charge density distribution calculations of Co NP on carbon and on N-doped carbon (**Figure S29**), where Co on N-doped carbon loses more electrons ($-0.483 e^-$), verifying the electron transfer from Co to N atoms.

Page 9, lines 8-19.

To further investigate the electronic structure and coordination environment of Co species in Co-N-CTS, Co K-edge X-ray absorption near-edge structure (XANES) spectroscopy and extended X-ray absorption fine structure (EXAFS) analysis were also conducted (**Figure 2g and 2h**)^{51,52}. For comparison, Co foil, CoO, Co_3O_4 , and Co-CTS were used as the reference samples. The Co K-edge in the XANES spectra shows that the absorption near edge spectrum for Co-N-CTS lies between those for Co foil and CoO (**Figure 2g**), indicating partial electron transfer from Co to N with a valence state between 0 and +2. However, the absorption near the

edge spectrum of Co-CTS is similar to that of Co foil, indicating that near 0 (metallic) Co lacks electron transfer between Co and the C substrate. Additionally, the Fourier transformed EXAFS (FT-EXAFS) spectra of the Co K-edge for Co-N-CTS further indicate the coexistence of Co-N/O and metallic Co-Co scattering paths (**Figure 2h**), whereas for Co-CTS, only a predominant peak at $\approx 2.16 \text{ \AA}$ (Co-Co bonds) was observed, which is typically assigned to the Co NPs⁵³⁻⁵⁵. Page 11, lines 19-21.

In addition, the strong metal-substrate interaction between Co NPs and N-doped carbon and their electron transfer significantly regulate the electronic structure of Co to optimize the binding energies toward intrinsically enhanced catalytic activity^{41,61}.

Updated Figure. 2:

Figure 2. Coordinated CTS synthesized high-density and ultrasmall nanoparticles on hierarchical porous carbon. (a) The porous structure of Co-N-CTS with a multitude of open pores. (b) AFM image of 2D porous carbon showing a thin film structure and a thickness of $< 4 \text{ nm}$. (c) N_2 sorption isotherms of Co-N-CTS and Co-N-furnace. (d) TEM image of Co-N-CTS with a size of $\sim 3.2 \text{ nm}$. (e) HAADF and EDS mapping images of Co-N-CTS. (f) High-resolution XPS deconvolution of the N element of Co-N-CTS-1000. (g) The normalized XANES spectra and (h) k^3 -weighted Fourier transform of EXAFS spectra at the Co K-edge of Co-N-CTS, Co-CTS, and reference samples.

Modification in revised Supporting Information:

In addition, the XPS spectrum and charge density distribution are also added to **Figures S28** and **S29** and **Tables S3** and **S4** of the Supporting Information as suggested by the reviewer.

4. the authors demonstrated that the Co-CTS ($326 \text{ m}^2 \text{ g}^{-1}$) has a higher Brunauer–Emmett–Teller (BET) specific surface area than the sample ($190 \text{ m}^2 \text{ g}^{-1}$) by the nitrogen adsorption and desorption method. These pores are critical for nanoparticle exposure and electrolyte penetration to largely improve performance in electrochemical reactions. Is there any pore size effect on electrocatalytic performance?

Response: We truly appreciate the reviewer’s valuable comment about the effects of pores on electrocatalytic performance. We agree with the reviewer that higher surface areas can expose more nanoparticles (active sites) and enable better electrolyte penetration, thus contributing to improved catalytic performances. Namely, there is a positive pore size effect that benefits the catalytic performance. Indeed, this is exactly the advantage of our coordinated CTS to enable the formation of open porous structures and 2D thin carbon supports, thus benefiting nanoparticle exposure, electrolyte penetration, and catalytic performance.

As shown in **Figure R14a**, Co-N-CTS obviously displays a much larger current density and faster kinetics than Co-N-furnace, particularly in the high potential region, which can be attributed to the unique hierarchical porous structure facilitating active site exposure, electrolyte penetration, and fast gas/mass transport.

Meanwhile, we also tried to eliminate the pore size effect and compare the intrinsic activity of Co-N-CTS by normalizing the activity by the ECSA. The ECSAs of Co-N-CTS and Co-N-furnace were measured, where the Co-N-CTS sample shows higher ECSA values ($2349 \text{ cm}^2 \text{ ECSA}$) than that of Co-N-furnace ($620 \text{ cm}^2 \text{ ECSA}$), which is consistent with the BET results and confirms that Co-N-CTS has a higher active surface area. We then show the ECSA-normalized LSV curves in **Figure R14b**. Nevertheless, the normalized specific activity of Co-N-CTS is higher than that of Co-N-furnace, indicating higher intrinsic activity in Co-N-CTS samples.

Figure R14. (a) LSV curves and (b) ECSA-normalized LSV curves of Co-N-CTS and Co-N-furnace.

Modification in the revised manuscript:

Following the reviewer’s valuable suggestion, to help the reader’s understanding of the pore size effect, we added more discussion in our revised manuscript.

Page 10, lines 19-21.

Additionally, Co-N-CTS displays a much larger current density in the high potential region, which can be attributed to the unique hierarchical porous structure facilitating mass transport.

Page 11, lines 1-4.

The intrinsic activities of Co-N-CTS are also investigated by normalizing the current based on the ECSA, where the ECSA-normalized specific activity of Co-N-CTS is still higher than that of Co-N-furnace, indicating higher intrinsic activity in Co-N-CTS samples (**Figure S33**).

Page 11, lines 21-23; page 12, line 1 and 2.

Last but not least, the catalysts are dispersed on open and hierarchical porous structures with large surface areas, which effectively exposes these active sites, facilitates electrolyte penetration, and benefits gas/mass transportation, all of which are extremely beneficial for electrocatalytic ORR and OER, especially at large current densities.

Modification in revised Supporting Information:

In addition, ECSA-normalized LSV curves are also added to **Figure S33** in the Supporting Information as suggested by the reviewer.

5. In Figure 3e, the Co-CTS with 2 Melm catalyst (280 mV to drive a current density of 10 mA cm⁻²) exhibited higher OER activity than Co-CTS without 2 Melm (330 mV to drive a current density of 10 mA cm⁻²). However, in the potential region from 1.5 V to 1.7 V, there is no significant difference in the OER activity of the two catalysts (with and without 2 Melm) because the reaction kinetics of Co-CTS with the 2 Melm catalyst slows down as the potential increases. Therefore, the authors should measure the reaction kinetics during the OER operation of prepared catalysts by calculating the Tafel slope.

Response: We truly appreciate the reviewer's comment about the OER activity and their reaction kinetics. First, to better display the difference in the OER activity at high potentials, we updated the figure drawing in the OER LSV curves to the 0-150 mA/cm² range and separately compared them.

As shown in **Figure R15**, in the low potential region, Co-N-CTS needs an overpotential of 280 mV to achieve a current density of 10 mA cm⁻², which is similar to that of commercial RuO₂ (280 mV) and far lower than that of Co-CTS (i.e., without 2-Melm) (330 mV) and Co-N-furnace (400 mV), indicating the excellent OER performance of Co-N-CTS. Particularly, in the high overpotential region, Co-N-CTS displays a much larger current density than commercial RuO₂ and other control samples, indicating much better kinetics at high potentials, which can be attributed to the open and hierarchical porous structure facilitating better nanoparticle/active site exposure, electrolyte penetration, and fast gas/mass transport, which is particularly important at high overpotentials.

Figure R15. LSV curves of commercial RuO₂, Co-N-furnace, Co-CTS, and Co-N-CTS.

According to the Reviewer's comment, we also analyzed the Tafel slope of the OER curves to compare the reaction kinetics. As shown in **Figure R16**, the Tafel slope of Co-N-CTS (94.5 mV dec⁻¹) is near that of commercial RuO₂ (77.5 mV dec⁻¹) and lower than that of Co-CTS (96.8 mV dec⁻¹) and Co-N-furnace (160.3 mV dec⁻¹), suggesting the fast OER kinetics of Co-N-CTS. Note that the Co-N-furnace has a much higher Tafel slope (slow kinetics), which is largely due to the buried Co nanoparticles inside the 3D pyrolyzed MOF, resulting in severe performance degradation. These results indicate that the excellent OER performance of Co-CTS comes from not only good intrinsic activity but also structure-benefited fast kinetics, particularly at high overpotentials.

Figure R16. Tafel slope of commercial RuO₂, Co-N-furnace, Co-CTS, and Co-N-CTS.

Modification in the revised manuscript:

Following the reviewer's valuable suggestion, to help the reader's understanding of the OER performance of the Co-N-CTS sample, we updated the OER LSV curves in **Figure 3** and added more detailed discussions in our revised manuscript.

Page 10, lines 16-23.

In terms of the oxygen evolution reaction (OER), Co-N-CTS also displays excellent OER performance even compared to noble RuO₂ catalysts (**Figure 3e**). The overpotential of Co-N-CTS and RuO₂ is ~280 mV, which is similar to that of commercial RuO₂ but much lower than that of the other control samples (Co-CTS or Co-N-furnace, **Figure 3f**). Additionally, Co-N-CTS displays a much larger current density in the high potential region, which can be attributed to the unique hierarchical porous structure facilitating mass transport. As shown in **Figure S32**, the Tafel slope of Co-N-CTS (94.5 mV dec⁻¹) is slightly larger than that of commercial RuO₂ (77.5 mV dec⁻¹) and lower than those of Co-CTS (96.8 mV dec⁻¹) and Co-N-furnace (160.3 mV dec⁻¹), suggesting the fast OER kinetics of Co-N-CTS.

Modification in revised Supporting Information:

In addition, Tafel slopes are also added to **Figure S32** in the Supporting Information as suggested by the reviewer.

6. Furthermore, stability is a critical factor to demonstrate the novelty of the prepared catalysts toward ORR and OER. The authors should conduct the stability test (chronopotentiometry or chronoamperometry test) or durability cycle (ADT test) to prove the stability of the Co-CTS catalyst. Moreover, the structural characterizations (TEM, XRD, XPS, etc.) for the catalyst after the stability test or durability cycle are needed to understand the change in the structural properties and chemical states of the catalyst surface and prove the stability of the prepared catalyst. This analysis might help explain why the ORR and OER performance was enhanced by Co-CTS with the 2 Melm catalyst. Therefore, the authors should provide durability or stability measurement results of prepared catalysts and additional characterizations after the durability or stability test with a detailed discussion about the change of crystal structure and oxidation structure after long-term operation.

Response: We truly appreciate the reviewer's comment about the stability and durability of the catalysts. In our original manuscript, for the ORR, accelerated potential cycling tests were conducted in the interval between 0.6 and 1.0 V (versus RHE) at a scanning rate of 50 mV s^{-1} for up to 5000 cycles. As displayed in **Figure R17a**, Co-N-CTS shows a negative shift of only 6 mV for the half-wave potential, indicating good ORR stability in accelerated electrochemical cycling. For the OER, Co-N-CTS also displayed excellent stability in the long-term OER (**Figure R17b**).

Figure R17. (a) The ORR-LSV curves before and after 5000 cycles and (b) the OER stability tests of Co-N-CTS.

Accordingly, we performed a new stability test and characterized the XRD, XPS, SEM, and TEM patterns after the OER and ORR stability tests.

OER and ORR stability: The long-term durability of the OER was also tested at a constant current density of 10 mA cm^{-2} (chronopotential test) for 20 h (**Figure R18a**). The Co-N-CTS catalyst maintained its initial potential after the test for 20 h and surpassed that of the noble RuO_2 catalyst.

In addition, after the current-time ($i-t$) chronoamperometry measurements at a constant potential of 0.6 V versus RHE for 10 h, the retention rate of current density for the Co-N-CTS

catalyst is 94.8 %, much higher than that of commercial Pt/C (77.4 %) (**Figure R18b**). These results indicate the excellent stability of the Co-N-CTS catalyst.

Figure R18. the OER and ORR stability tests of Co-N-CTS, commercial RuO₂ (OER), and commercial Pt/C (ORR).

Post-analysis. We performed the poststructure analysis for Co-N-CTS catalysts after the long-time stability test. As shown in **Figure R19**, the Co-N-CTS catalysts still maintained an open and hierarchical porous structure, high-density Co nanoparticles were still uniformly dispersed on the 2D porous carbon, and no obvious agglomeration was found, indicating the good structural stability of our catalysts.

Figure R19. SEM and TEM images of Co-N-CTS after 20 h OER (a-c) and 10 h ORR (d-f) tests.

While the XRD patterns show an overall metallic Co phase, the XPS spectra reveal that metallic Co still exists, and a more oxidized surface is present after the stability test of ORR and OER, which is commonly observed for nonnoble metals (**Figure R20**) (*Appl. Catal. B Environ.* 2021, 281, 119514, *Appl. Catal. B Environ.* 2020, 279, 119407, *Adv. Mater.* 2019, 31, 1901666).

Figure R20. XRD and Co XPS spectra of Co-N-CTS after 20 h OER and 10 h ORR tests.

In summary, the Co-N-CTS catalysts still maintained the open and hierarchical porous structure, high-density Co nanoparticles were still uniformly dispersed on the 2D porous carbon, and no obvious agglomeration was found, indicating excellent structural stability in the Co-N-CTS catalysts. Additionally, the XRD and XPS results showed an overall metallic Co phase with an increasing oxidized surface after the stability test of ORR and OER, which is commonly observed for non-noble metals. These results indicate good structural and performance stability of the Co-CTS catalyst.

Modification in the revised manuscript:

Following the reviewer’s valuable suggestion, we added more detailed discussions in our revised manuscript.

Page 10, lines 12-15.

In addition, after the current-time (*i-t*) chronoamperometry measurements at a constant potential of 0.6 V versus RHE for 10 h, the retention rate of current density for the Co-N-CTS catalyst is 94.8 %, higher than that of commercial Pt/C (77.4 %) (**Figure S31**), further verifying the excellent ORR stability of the Co-N-CTS catalyst.

Page 11, lines 4-7.

The long-term durability of the OER was also tested at a constant current density of 10 mA cm⁻² (chronopotential test) for 20 h (**Figure S34**). The Co-N-CTS catalyst maintained its initial potential after 20 h and surpassed that of the noble RuO₂ catalyst, indicating excellent OER stability.

Page 11, lines 8-14.

We also performed poststructure analysis for Co-N-CTS catalysts after the long-term stability test. The Co-N-CTS catalysts still maintained the open and hierarchical porous structure with high-density Co NPs uniformly dispersed on the 2D porous carbon, and no obvious agglomeration was found (**Figure S35**). Additionally, the XRD and XPS results showed a metallic Co phase with an increasing oxidized surface after the stability test of ORR and OER (**Figure S36**), which is commonly observed for nonnoble metals⁵⁶⁻⁵⁸. These results validate the good structural and performance stability of the Co-N-CTS catalyst.

Modification in revised Supporting Information:

In addition, the XRD, XPS, SEM, and TEM results after the OER and ORR stability tests are also added to **Figure S31** and **Figure S34-36** of the Supporting Information as suggested by the reviewer.

Reviewer#2:

In this work, Shi et al. reported the synthesis of high-density of ultrasmall nanoparticles by carbothermic shock. Metal salts and C₂H₈N₂ were mixed and loaded onto carbon black. The rapid Joule heating pyrolyzed the precursors and nanoparticles were formed. The main conclusion claimed in this work is the high density and ultrasmall size of the obtained nanoparticles. They authors demonstrated the application as bifunctional oxygen electrocatalysts. The main concern from the reviewer is that the carbothermic shock synthesis of similar metal nanoparticles is very well documented. The novelty of this work might not be warranted to publish in a Nature series journal.

Response: We sincerely appreciate the reviewer for the valuable comments, which have made a tremendous improvement to our manuscript. We have tried our best to address these issues and revised the manuscript in a point-by-point manner. We hope that these revisions address all the concerns you raised and make the manuscript significantly stronger. We truly thank you for your time and attention.

1. Tremendous works were published recently to use the carbothermic shock synthesis of single-metal nanoparticles, bimetallic nanoparticles, high-entropy nanoparticles, etc. The main conclusion of this work is the high density and small size. However, with regard to high density, Song et al. reported the generation of high-density nanoparticles in the carbothermal shock method (Sci. Adv. 7, eabk2984 (2021)), with a similar strategy to enhance the metal-carbon substrate interaction. They achieved a coverage of 85%, which is higher than this work is showing. With regard to ultrasmall size, Lacey et al. reported the multimetallic nanoparticle size of 2-3 nm by the carbothermic shock method (Nano Lett. 2019, 19, 8, 5149–5158), which is smaller than the values the authors reported here (3.2 nm – 22 nm, Figure 5). The same strategy of rapid Joule heating pyrolysis of MOF for nanoparticles synthesis has already reported by the authors (Nano Energy, 2022, doi: 10.1016/j.nanoen.2022.107125). In that work, they reported -sub-3 nm Co nanoparticle with high loading up to 41%, where the particles size is, again, smaller than the values reported here. Hence, I cannot see substantial advance of this work over previous publications.

Response: We sincerely appreciate the reviewer's comment. We thank the reviewer for pointing out that the carbothermal shock (CTS) method is now a favorable tool used to synthesize a wide range of nanoparticles/catalysts because of its superiority in precise temperature control and high efficiency. In fact, many of the coauthors in this paper have been working on CTS synthesis for a long time and contributed significantly.

We truly understand the reviewer's concern about the novelty of the reported coordinated CTS, and we would like to take this opportunity to clearly compare our work with previous important literature. In the meantime, we disagree with the reviewer that *"The main conclusion of this work is the high density and small size."* Our work first introduced **the "metal-ligand precursor design (i.e., coordination chemistry)"** into catalyst synthesis by CTS, during which the metal-ligand precursor (e.g., Co²⁺ and dimethylimidazole C₅H₈N₂) undergoes **in situ assembly into locally ordered porous structures with strong coordination (N→Co²⁺)**, resulting in **not only ultrasmall and high-density Co NPs stabilized by strong metal-**

substrate interactions (Co-N bonds) but also open and hierarchical porous structures and 2D thin carbon supports as a result of explosively pyrolyzing the in situ assembled porous frameworks, which has never been reported before. The combination of coordination chemistry and CTS pyrolysis leads to the general and versatile synthesis of well-dispersed NPs stabilized on hierarchical porous carbon films, which we found is crucial in catalysis because of the significantly increased active site density, effective catalyst exposure and enhanced mass transport, as well as intrinsically improved catalysis due to strong metal-support interactions.

Below is a detailed comparison of important CTS works:

1. Conventional CTS (metal precursor + rapid heating): Yao et al. first reported CTS technology for synthesizing a wide range of single- to multicomponent metal nanoparticles (*Science* 2018, 359, 1489-1494). CTS is achieved by loading metal salts on a conductive carbon substrate and then initiating transient high-temperature heating by passing a pulsed electric current through the carbon substrate (**Figure R21**).

CTS itself can synthesize small nanoparticles by substantially limiting the heating duration (i.e., to 55 ms). Nevertheless, particle agglomeration and sintering are inevitable at high temperatures, typically resulting in > 10-20 nm nanoparticles (particularly for nonnoble metals) by conventional CTS. In addition, these nanoparticles are only affiliated/anchored on the carbon substrate, showing a relatively low surface dispersion and utilization.

Figure R21. Schematic of the conventional CTS process. (Introduced by *Nature outlook*, <https://www.nature.com/articles/d41586-021-01665-w>).

2. Enhanced CTS (introducing strong metal-substrate interactions): To enable the synthesis of ultrasmall and high-density nanoparticles via CTS, enhanced metal-substrate interactions

have been explored. The introduction of defects and anchoring sites (e.g., S, N, P, etc.) on the substrate can effectively stabilize nanoparticles by forming metal-substrate bonds, which promotes the synthesis of ultrasmall nanoparticles (**Figure R22**).

Yao et al. reported the synthesis of Pt nanoclusters (~4.8 nm with high density (40 wt%) and even ~0.8 nm with low density) via CTS of defective carbon substrates (CO₂ activated CNF) (*ACS Appl. Mater. Interfaces* 2019, 11, 33, 29773). Additionally, as the reviewer noted, Lacey et al. reported the synthesis of multimetallic nanoparticles of < 5 nm in three-dimensionally ordered mesoporous (3DOM) carbon, which have not only sufficient defects but also ordered mesopores (~20 nm) to confine the growth of nanoparticles (*Nano Lett.* 2019, 19, 8, 5149), yet the nanoparticles are mainly limited to noble Ru-based catalysts with a sporadic distribution (low density due to requiring pores for confinement). Importantly, Song et al. reported the generation of high-density nanoparticles by introducing partially carbonized cellulose (highly defective) as the substrate in the CTS process (*Sci. Adv.* 7, eabk2984), which leads to an astonishingly high surface area coverage of >85 % by maximizing the defect sites of cellulose (yet with a relatively large particle size of ~35 nm).

Figure R22. Schematic image of the enhanced CTS process by introducing strong metal-substrate interactions, which can achieve ultrasmall NPs due to effective anchoring on the substrate.

3. Our study, Coordinated CTS (precursor design and in situ assembly):

The above approaches are all effective in obtaining small and/or high-density nanoparticles due to effective anchoring of the specially treated substrate (i.e., enhanced metal-substrate interaction). However, none of these studies have shown using flexible precursor design (i.e., coordination chemistry) to achieve not only small and high-density nanoparticles but also (1) open and hierarchical porous structures and (2) porous and 2D thin film carbon supports that are essential for catalyst exposure (activity) and mass transport (kinetics), especially at high overpotentials (**Figure R23**). These unique structural features are only formed by explosive

pyrolysis of the in situ assembled, locally ordered porous frameworks as a result of metal-ligand coordination (main text **Figure 4**), which has not been reported by any previous works. In addition, considering the huge composition flexibility in coordination chemistry design, the coordinated CTS process can enable the general synthesis of versatile catalysts with ultrasmall size and porous structures, even beyond crystalline MOF chemistry (main text **Figure 5**), without any special requirement for the substrate, thus significantly improving the generality and versatility of catalyst synthesis.

Figure R23. (a) Schematic of our coordinated CTS process, showing the in situ assembly between metal and ligands and the formation of not only high-density nanoparticles but also (b) open and porous structures, as shown in the SEM, AFM, and TEM results.

As the reviewer noted, our recent paper reported the rapid Joule heating pyrolysis of MOF for nanoparticle synthesis (*Nano Energy* 2022, 97, 107125). However, these two works are significantly different in the following aspects (**Figure R24**):

(1) That work utilized MOF, an already ordered crystalline structure, for CTS pyrolysis. Although one can obtain high-density and small nanoparticles due to ligand coordination, stable MOF pyrolysis will lead to **only a dense and compact structure with nanoparticles mostly buried inside**, which is drastically different from the in situ metal-ligand coordination and formation of locally ordered porous frameworks that eventually leads to significantly tunable pore morphology and hierarchy (micronano pores).

(2) As a result, MOF-derived catalysts show **inferior catalytic performances due to the buried nanoparticles and a compact structure**, while our catalysts displayed superior catalytic performances that are among the best Co electrocatalysts. This is because the hierarchical porous structure with a multitude of pores is essential for exposing active sites, enabling electrolyte penetration, and facilitating mass transport for improved catalytic performance (activity and kinetics).

(3) Importantly, MOF pyrolysis **first** requires **MOF synthesis, which is not only time-consuming (~days) but also limited to only MOF compositions**. In contrast, the current work

starts with metal-ligand precursor design combined with CTS synthesis (~ 1 s), which is not only extremely efficient but also general and versatile to all types of coordination networks, particularly beyond MOF chemistry.

Figure R24. Comparison between (a) CTS pyrolysis of MOF and (b) coordinated CTS pyrolysis of metal-ligand precursors and their surface areas and electrochemical performances.

In summary, our current paper is significantly different from previous CTS processes, as it is the first to introduce a **metal-ligand precursor design to induce in situ coordination and porous framework formation** to achieve **not only** high density and small nanoparticles **but also** open and hierarchical porous structures and 2D thin porous supports that are desirable for high surface area dispersion, catalyst exposure, electrolyte penetration, and mass transport, which is the origin of the superior electrocatalytic performance of our samples. Considering the huge design space and flexibility in coordination chemistry, the introduction of precursor design in the CTS process significantly improves the generality and versatility of catalyst synthesis.

We also compared the different methods side-by-side in **Table R4**.

Table R4. Comparison of various CTS methods.

	1.Substrate	Special substrate?	2.Precursor	Precursor design?	Compositions	In situ assembly / tunable pores?	3. Particles	4.Hierarchical porosity?	Reference
Conventional CTS	Carbon	X	Metal salt (e.g., CoCl ₂)	X	None specific	X	~10-20 nm	X	Science 2018, 359, 1489-1494 (our previous work)
Enhanced CTS (enhancing the metal-substrate interaction)	CO ₂ activated carbon (defective carbon)	√	Metal salt (e.g., H ₂ PtCl ₆)	X	None specific	X	~5 nm (high density) ~0.8 nm (low density)	X	ACS AMI 2019, 11, 29773 (our previous work)
	3D mesoporous carbon (defective, pore confinement)	√	Metal salt (e.g., RuCl ₂)	X	None specific	X	~ 3 nm (low density)	X (only mesopores)	Nano Lett. 2019, 19, 8, 5149-5158 (Lacet et. al.)
	Cellulose (maximized defective carbon)	√	Metal salt (e.g., CuCl ₂)	X	None specific	X	~35 nm (High density coverage > 85%)	X	Sci. Adv. 2021, 7, eabk2984 (Song et. al.)
Coordinated CTS (coordinated metal-ligands in precursor design)	Carbon (carbon for CTS)	X	Crystalline MOF (e.g., ZIF-67)	√	Only MOF	X	< 3 nm (high density)	X (micropores < 2 nm)	Nano Energy 2022, 97, 107125
	Carbon (carbon for CTS)	X	Metal + ligands (e.g. Co ₂ + & 2Melm)	√	Coordination chemistry (Beyond MOF)	√	~ 3.2 nm (high density)	√ open/hierarchical pores (μm-nm)	This work

Modification in the revised manuscript:

Following the reviewer's valuable suggestion, we added more detailed discussions in our revised manuscript.

We added "on Hierarchical Porous Carbon" to our title as "*Transient and General Synthesis of High-density and Ultrasmall Nanoparticles on Hierarchical Porous Carbon via Coordinated Carbothermal Shock*" to highlight that coordinated CTS could enable the general and versatile synthesis of not only high-density and ultrasmall nanoparticles because of coordination confinement but also the hierarchical porous carbon support because of pyrolyzing the in situ assembled porous frameworks, which is significantly beneficial for catalyst exposure and mass transport.

Page 3, lines 13-23.

Recently, carbothermal shock (CTS) technology has emerged as a promising methods for synthesizing a wide range of catalysts (size and composition) due to its precise heating control and extreme efficiency (in seconds)^{4,20-22}. However, particle agglomeration and sintering are inevitable at high temperatures, typically resulting in larger nanoparticles (> 10 nm, for nonnoble metals). To enable the synthesis of ultrasmall NPs, enhanced metal-substrate interactions have been explored by introducing highly defective substrates or heteroatom doping (e.g., S, N, P) as anchoring sites for effective dispersion and/or high surface coverage²³⁻²⁵. However, these methods often require special substrate treatments or strict bond formation, while the formed NPs are only anchored on the backbone substrate, leading to relatively complex syntheses and/or low surface dispersion. A comparison of various CTS methods is summarized in **Table S1**.

Page 4, lines 5-9.

Previously, rapid CTS pyrolysis was used for MOF microcubes to synthesize ultrasmall NPs (~ 4 nm)³³, yet most NPs were found to be buried inside the carbon matrix with limited pore sizes (< 2 nm), thus causing serious metal deactivation and substantial resistance in mass transfer (electrolyte and gas) along with inferior catalytic performances.

Page 4, lines 15-18.

Therefore, a more general and controllable method is highly desirable to achieve not only ultrasmall and high-density NPs but also open porous structures that are essential for catalyst exposure and efficient mass transport, thus significantly improving the versatility of catalyst

synthesis and high-performance catalysis.

Modification in the supporting information:

The comparison among various CTS is included in **Table S1** in the Supporting Information.

2. The authors used the Co nanoparticles in ORR and OER. Co nanoparticles as electrocatalysts have been widely reported. The authors should compare their materials performances with literature values and present that for a comparison for the readers to see. Does the high density and small size enhance the performance?

Response: We truly appreciate the reviewer’s comment about the comparison between the Co-N-CTS catalyst and the widely reported Co catalysts. Owing to the small size and high density, and particularly the hierarchical porous structure, our Co-N-CTS sample exhibited outstanding ORR and OER activity, which is superior or at least comparable to most of the reported state-of-the-art nonprecious metal catalysts (**Figure R25** and **Table R5**). In particular, our catalysts demonstrated the lowest potential window for bifunctional ORR/OER at ~0.65 V, which is comparably smaller than that of the other catalysts.

Figure R25. Comparison of the bifunctional oxygen electrocatalytic performances ($E_{1/2}$ in ORR and $E_{j=10}$ in OER). The marked data are the potential difference between OER ($E_{j=10}$) and ORR ($E_{1/2}$) ($\Delta E = E_{j=10} - E_{1/2}$) between Co-CTS and other reported Co-based bifunctional electrocatalysts in the literature.

Table R5. Performance comparison in ORR and OER between Co-N-CTS and other reported Co-based bifunctional electrocatalysts in the literature.

Catalysts	ORR $E_{1/2}$ (V)	OER η_{10} (mV)	ΔE (V)	Reference
Co-N-CTS	0.86	280	0.65	This work
OLC/Co-N-C	0.86	350	0.72	Angew. Chem., Int. Ed. 2021, 60, 12759.
NCNTFs	0.87	370	0.73	Nat. Energy. 2016,1, 15006.
CNT@SAC-Co/NCP	0.87	380	0.74	Adv. Funct. Mater. 2021, 31, 2103360.
Co₃O_{4-x} HoNPs @HPNCS	0.83	340	0.74	Angew. Chem. Int. Ed. 2019, 58, 13840-13844.

Co@hNCTs-800	0.87	400	0.76	Nano Energy 2020, 71, 2211.
Co SA/NCFs	0.85	380	0.76	Nano Lett. 2022, 22, 2497–2505.
Co-NCNT	0.86	400	0.77	Energy Storage Mater. 2019, 20, 234-242.
Co/CNWs/CNFs	0.82	410	0.82	Adv. Funct. Mater. 2021, 31, 2105021.
Co-POC	0.83	470	0.87	Adv. Mater. 2019, 31, 1900592.
CoSAs-NGST	0.89	560	0.9	Adv. Funct. Mater. 2021, 31, 2010472.

Modification in the revised manuscript:

Following the reviewer’s valuable suggestion, to help the reader’s understanding of the comparison between the Co-N-CTS catalyst and the widely reported Co nanoparticle electrocatalysts, we added more detailed discussions in our revised manuscript.

Page 11, lines 15-17.

The outstanding ORR and OER activity of our catalyst is comparable or superior to that of most of the reported state-of-the-art nonprecious metal catalysts (**Figure S37** and **Table S5**), which could be attributed to the following reasons.

Modification in revised Supporting Information:

In addition, the electrocatalytic performance comparison is also added to **Figure S37** and **Table S5** in the Supporting Information as suggested by the reviewer.

3. The measurement conditions of TGA (Fig. 4a, Fig. S23) should be provided. Was it measured in inert gas or air? What is the heating rate? The y-axis values are missing for these TGA plots.

Response: We appreciate the reviewer’s comment. The measurement conditions of TGA have been updated in the Methods section. It was measured at a heating rate of 20 °C/min in Ar gas. Additionally, the y-axis values of these TGA plots are also added (**Figure R26** and **R27**).

Figure R26. TGA-DTG profile of MOC from 50 to 1000 °C in Ar gas.

Figure R27. TGA-DTG profiles of (a) 2-Melm and (b) $\text{Co}(\text{NO}_3)_2$ from 50 to 600 °C in Ar gas.

Modification in the revised manuscript:

Following the reviewer’s valuable suggestion, we updated the TGA-DSC profile in **Figure 4**. Page 19, lines 2 and 3.

Thermogravimetric analysis (TGA) was performed by Diamond TG/DTA (PerkinElmer Instruments) at a heating rate of 20 °C/min in Ar gas.

Modification in revised Supporting Information:

In addition, the updated TGA-DTG profile has also been added to **Figure S38** in the Supporting Information as suggested by the reviewer.

4. The authors show the synthesis at different heating rates of 0.1, 100, 1000, 10^4 °C/s. How is the heating rate controlled by the CTS process?

Response: We appreciate the reviewer’s comment on the heating rate controlled by the CTS process. In our CTS process, high-temperature heating is triggered by electrified Joule heating of the carbon substrate. Therefore, the temperature and heating rates can be adjusted by programming the electrical current. In our heating rate experiments, a heating rate of 10^4 °C/s is achieved by an instantly pulsing current from 0 to (20 V, 10 A), which is the highest heating achieved by the current system and setup. Other slower heating rates (100 and 1000 °C/s) can be achieved by programming the current stepwise at rates of 1 and 10 A/s. Finally, a heating rate of 0.1 °C/s is obtained by conventional furnace treatment (6 °C/min).

Modification in the revised manuscript:

Following the reviewer’s valuable suggestion, the method of the heating rate controlled by the CTS process has been added to the **Methods** section of our revised manuscript. Page 18, lines 15-19.

The heating rate of 10^4 °C/s is achieved by an instantly pulsing current to (20 V, 10 A), which is the highest heating achieved by the current system and setup. Other slower heating rates (100 and 1000 °C/s) can be achieved by programming the current stepwise at rates of 1 and 10 A/s. Finally, a heating rate of 0.1 °C/s is obtained by conventional furnace treatment (6 °C/min).

5. What is the chemical state of Co? Is it Co(0) or with some oxide composition? XPS or other analysis should be provided. How does the difference affect the electrocatalytic performance?

Response: We thank the reviewer for the comment. Since **Reviewer #1** also asked about the state of Co species in Co-N-CTS (**reviewer#1: comment 3**), please refer to that question for a detailed discussion. Briefly, as shown in **Figure R28**, a strong Co^0 signal was detected at 778.5 eV, suggesting the existence of metal Co NPs, while weak Co^{2+} and Co^{3+} signals were also observed, indicating surface oxidation of Co NPs, which are commonly found in nonnoble metals (*Angew. Chemie*, 2019, 131, 2648-2652, *Nano Energy* 2018, 52, 485-493).

Figure R28. The Co 2p XPS spectra of Co-N-CTS and Co-CTS.

We also studied the Co K-edge X-ray absorption near-edge structure (XANES) spectra and the extended X-ray absorption fine structure (EXAFS). The Co K-edge in the XANES spectra (**Figure R29a**) shows that the absorption near edge spectrum for Co-N-CTS lies between Co foil and CoO, indicating that the valence state of the Co species is between 0 and +2. However, the absorption near edge spectrum of Co-CTS lay near that for Co foil, indicating near 0 (metallic) Co. Additionally, the Fourier transformed EXAFS (FT-EXAFS) spectra of the Co K-edge for Co-N-CTS further indicate the coexistence of Co-N/O and metallic Co-Co scattering paths, respectively (**Figure R29b**), whereas for Co-CTS, only a predominant peak at ≈ 2.16 Å (Co-Co bonds) was observed, which is typically assigned to the Co NPs.

Figure R29. (a) Normalized XANES spectra at the Co K-edge of Co-N-CTS, Co-CTS, and reference samples. The inset shows the enlarged marked area. (b) k^3 -weighted Fourier transform of EXAFS spectra at the Co K-edge of Co-N-CTS, Co-CTS, and reference samples.

From the above analysis, it is clear that the Co species in the Co-CTS and the Co-N-CTS samples have significantly different electronic structures and coordination environments.

Therefore, we compared the catalytic performances of these two samples to illustrate the catalytic differences caused by the different Co species in the Co-CTS and Co-N-CTS samples (**Figure R30**). Obviously, Co-N-CTS exhibited better ORR and OER performances than the Co-CTS sample, indicating that the strong interaction between Co NPs and N-doped carbon and their electron transfer significantly regulate the electronic structure of Co to optimize the binding energies toward intrinsically enhanced catalytic activity (*Angew. Chemie Int. Ed.* 2019, 58, 12540-12544, *ACS Catal.* 2020, 10, 12376-12384).

Figure R30. Comparison of the (a) ORR and (b) OER performances of the Co-CTS and Co-N-CTS samples.

Modification in the revised manuscript:

Following the reviewer's valuable suggestion, we updated the XPS and XAS data and added more detailed discussions in our revised manuscript.

Page 8, lines 21-23; page 9, line 1 and 7.

In addition, by detailed analysis of the Co XPS spectra (**Figure S28, Tables S3 and S4**), a strong Co^0 signal was detected at 778.5 eV, while weak Co^{2+} and Co^{3+} signals were also observed, suggesting surface oxidation of Co NPs^{48,49}. Additionally, the peak position of metallic Co 2p shifts to a higher binding energy (~ 0.3 eV) in our Co-N-CTS sample, indicating partial electron transfer from Co to N and effective modulation of the electronic structure in Co-N-CTS by the strong metal-substrate interaction between Co NPs and N-doped carbon⁵⁰. The metal-substrate interaction and electron transfer are further verified by charge density distribution calculations of Co NP on carbon and on N-doped carbon (**Figure S29**), where Co on N-doped carbon loses more electrons (-0.483 eV), verifying the electron transfer from Co to N atoms.

Page 9, lines 8-19.

To further investigate the electronic structure and coordination environment of Co species in Co-N-CTS, Co K-edge X-ray absorption near-edge structure (XANES) spectroscopy and extended X-ray absorption fine structure (EXAFS) analysis were also conducted (**Figure 2g and 2h**)^{51,52}. For comparison, Co foil, CoO, Co_3O_4 , and Co-CTS were used as the reference samples. The Co K-edge in the XANES spectra shows that the absorption near edge spectrum for Co-N-CTS lies between those for Co foil and CoO (**Figure 2g**), indicating partial electron transfer from Co to N with a valence state between 0 and +2. However, the absorption near the edge spectrum of Co-CTS is similar to that of Co foil, indicating that near 0 (metallic) Co lacks electron transfer between Co and the C substrate. Additionally, the Fourier transformed EXAFS (FT-EXAFS) spectra of the Co K-edge for Co-N-CTS further indicate the coexistence of Co-N/O and metallic Co-Co scattering paths (**Figure 2h**), whereas for Co-CTS, only a predominant peak at ≈ 2.16 Å (Co-Co bonds) was observed, which is typically assigned to the Co NPs⁵³⁻⁵⁵.

Page 11, lines 19-21.

In addition, the strong metal-substrate interaction between Co NPs and N-doped carbon and their electron transfer significantly regulate the electronic structure of Co to optimize the binding energies toward intrinsically enhanced catalytic activity^{41,61}.

Modification in revised Supporting Information:

In addition, the XPS spectrum and charge density distribution are also added to **Figures S28** and **S29** and **Tables S3** and **S4** of the Supporting Information as suggested by the reviewer.

Reviewer #3 (Remarks to the Author):

In this manuscript, the authors report a new synthesis method of high-density nanoparticles (NPs) on 2D porous carbon based on combination of coordination chemistry design and carbo-thermal shock (CTS). The prepared Co NPs in this manuscript exhibit ultrasmall NPs with high density and show high OER, ORR and Zn-air performances. The concept of coordination chemistry for the generation of high-density NPs using CTS is proven by matching other metals (e.g., Cu) and ligands (BTC). Overall, this manuscript possesses novelty to some extent especially in CTS method, and it is well organized with comprehensive characterizations. Therefore, this manuscript can be accepted after addressing the following issues:

Response: We thank the reviewer for carefully reading our paper and appreciating our work as “*a new synthesis method of high-density nanoparticles (NPs) on 2D porous carbon based on combination of coordination chemistry design and carbo-thermal shock (CTS)*” and “*possesses novelty to some extent especially in CTS method*”. We appreciate the reviewer for the valuable comments, which have made a tremendous improvement to our manuscript. We truly thank you for your time and attention.

1. To arouse a broad interest from readership in this field, some important and closely related literature about recent progress of generation of high density NPs using CTS should be cited in the revised manuscript (e.g. Sci. Adv. 2021, 7, eabk2984; ACS Appl. Mater. Interfaces 2019, 11, 29773; Nat. Commun. 2020 11, 6373). Compared to previous studies, the originality and advancement of this manuscript should be further elaborated in the Introduction.

Response: We appreciate the reviewer’s insightful comment. The suggested references are all important works on high-density and ultrasmall nanoparticles. We have accordingly cited these references in the revised **introduction** section. In addition, we made a detailed comparison among various CTS methods to elaborate on the novelty and significance of the coordinated CTS.

Our work first introduced the “metal-ligand precursor design (i.e., coordination chemistry)” into catalyst synthesis by CTS, during which the metal-ligand precursor (e.g., Co^{2+} and dimethylimidazole $\text{C}_5\text{H}_8\text{N}_2$) undergoes in situ assembly into locally ordered porous structures with strong coordination ($\text{N} \rightarrow \text{Co}^{2+}$), resulting in not only ultrasmall and high-density Co NPs stabilized by strong metal-substrate interactions (Co-N bonds) but also open and hierarchical porous structures and 2D thin carbon supports as a result of explosively pyrolyzing the in situ assembled porous frameworks. The combination of coordination chemistry and CTS pyrolysis leads to the general and versatile synthesis of well-dispersed NPs stabilized on hierarchical porous carbon films, which we found is crucial in catalysis because of the significantly increased active site density, effective catalyst exposure and enhanced mass transport, as well as intrinsically improved catalysis due to strong metal-support interactions.

Below is a detailed comparison of important CTS works:

1. Conventional CTS (metal precursor + rapid heating): Yao et al. first reported CTS technology for synthesizing a wide range of single- to multicomponent metal nanoparticles (*Science* 2018, 359, 1489-1494). CTS is achieved by loading metal salts on a conductive carbon

substrate and then initiating transient high-temperature heating by passing a pulsed electric current through the carbon substrate.

CTS itself can synthesize small nanoparticles by substantially limiting the heating duration (i.e., to 55 ms). Nevertheless, particle agglomeration and sintering are inevitable at high temperatures, typically resulting in > 10-20 nm nanoparticles (particularly for nonnoble metals) by conventional CTS. In addition, these nanoparticles are only affiliated/anchored on the carbon substrate, showing a relatively low surface dispersion and utilization.

2. Enhanced CTS (introducing strong metal-substrate interactions): To enable the synthesis of ultrasmall and high-density nanoparticles via CTS, enhanced metal-substrate interactions have been explored. The introduction of defects and anchoring sites (e.g., S, N, P, etc.) on the substrate can effectively stabilize nanoparticles by forming metal-substrate bonds, which promotes the synthesis of ultrasmall nanoparticles.

Yao et al. reported the synthesis of Pt nanoclusters (~4.8 nm with high density (40 wt%) and even ~0.8 nm with low density) via CTS of defective carbon substrates (CO₂ activated CNF) (*ACS Appl. Mater. Interfaces* 2019, 11, 33, 29773). Additionally, as the reviewer noted, Lacey et al. reported the synthesis of multimetallic nanoparticles of < 5 nm in three-dimensionally ordered mesoporous (3DOM) carbon, which have not only sufficient defects but also ordered mesopores (~20 nm) to confine the growth of nanoparticles (*Nano Lett.* 2019, 19, 8, 5149), yet the nanoparticles are mainly limited to noble Ru-based catalysts with a sporadic distribution (low density due to requiring pores for confinement). Importantly, Song et al. reported the generation of high-density nanoparticles by introducing partially carbonized cellulose (highly defective) as the substrate in the CTS process (*Sci. Adv.* 7, eabk2984), which leads to an astonishingly high surface area coverage of >85 % by maximizing the defect sites of cellulose (yet with a relatively large particle size of ~ 35 nm).

3. Our study, Coordinated CTS (precursor design and in situ assembly):

The above approaches are all effective in obtaining small and/or high-density nanoparticles due to effective anchoring of the specially treated substrate (i.e., enhanced metal-substrate interaction). However, none of these studies have shown using flexible precursor design (i.e., coordination chemistry) to achieve not only small and high-density nanoparticles but also (1) open and hierarchical porous structures and (2) porous and 2D thin film carbon supports that are essential for catalyst exposure (activity) and mass transport (kinetics), especially at high overpotentials. These unique structural features are only formed by explosive pyrolysis of the in situ assembled, locally ordered porous frameworks as a result of metal-ligand coordination (main text **Figure 4**), which has not been reported by any previous works. In addition, considering the huge composition flexibility in coordination chemistry design, the coordinated CTS process can enable the general synthesis of versatile catalysts with ultrasmall size and porous structures, even beyond crystalline MOF chemistry (main text **Figure 5**), without any special requirement for the substrate, thus significantly improving the generality and versatility of catalyst synthesis.

In summary, our current paper is significantly different from previous CTS processes, as it is the first to introduce a **metal-ligand precursor design to induce in situ coordination and**

porous framework formation to achieve **not only** high density and small nanoparticles **but also** open and hierarchical porous structures and 2D thin porous supports that are desirable for high surface area dispersion, catalyst exposure, electrolyte penetration, and mass transport, which is the origin of the superior electrocatalytic performance of our samples. Considering the huge design space and flexibility in coordination chemistry, the introduction of precursor design in the CTS process significantly improves the generality and versatility of catalyst synthesis.

Modification in the revised manuscript:

Following the reviewer's valuable suggestion, we added more detailed discussions in our revised manuscript.

Page 3, lines 13-23.

Recently, carbothermal shock (CTS) technology has emerged as a promising methods for synthesizing a wide range of catalysts (size and composition) due to its precise heating control and extreme efficiency (in seconds)^{4,20-22}. However, particle agglomeration and sintering are inevitable at high temperatures, typically resulting in larger nanoparticles (> 10 nm, for nonnoble metals). To enable the synthesis of ultrasmall NPs, enhanced metal-substrate interactions have been explored by introducing highly defective substrates or heteroatom doping (e.g., S, N, P) as anchoring sites for effective dispersion and/or high surface coverage²³⁻²⁵. However, these methods often require special substrate treatments or strict bond formation, while the formed NPs are only anchored on the backbone substrate, leading to relatively complex syntheses and/or low surface dispersion. A comparison of various CTS methods is summarized in **Table S1**.

Page 4, lines 5-9.

Previously, rapid CTS pyrolysis was used for MOF microcubes to synthesize ultrasmall NPs (~ 4 nm)³³, yet most NPs were found to be buried inside the carbon matrix with limited pore sizes (< 2 nm), thus causing serious metal deactivation and substantial resistance in mass transfer (electrolyte and gas) along with inferior catalytic performances.

Page 4, lines 15-18.

Therefore, a more general and controllable method is highly desirable to achieve not only ultrasmall and high-density NPs but also open porous structures that are essential for catalyst exposure and efficient mass transport, thus significantly improving the versatility of catalyst synthesis and high-performance catalysis.

Modification in the supporting information:

The comparison among various CTS methods is included in **Table S1** in the Supporting Information.

2. The authors should represent morphology of Co NPs (e.g. SEM and/or TEM images) after electrochemical stability test.

Response: We appreciate the reviewer's comment. Since **Reviewer #1** also asked about the stability and durability of the catalysts (**reviewer#1: comment 6**), please refer to that question for a detailed discussion.

Briefly, we performed the poststructure analysis for Co-N-CTS catalysts after the long-time stability test. As shown in **Figure R31**, the Co-N-CTS catalysts still maintained an open and hierarchical porous structure, high-density Co nanoparticles were still uniformly dispersed on the 2D porous carbon, and no obvious agglomeration was found, indicating the good structural stability of our catalysts. While the XRD patterns show an overall metallic Co phase, the XPS spectra reveal that metallic Co still exists, and a more oxidized surface is present after the stability test, which is commonly observed for nonnoble metals (**Figure R32**).

Figure R31. SEM and TEM images of Co-N-CTS after 20 h OER (a-c) and 10 h ORR (d-f) tests.

Figure R32. XRD and Co XPS spectra of Co-N-CTS after 20 h OER and 10 h ORR tests.

In summary, the Co-N-CTS catalysts maintained the open and hierarchical porous structure, high-density Co nanoparticles were still uniformly dispersed on the 2D porous carbon, and no obvious agglomeration was found, indicating excellent structural stability in the Co-N-CTS catalysts.

Modification in the revised manuscript:

Following the reviewer’s valuable suggestion, we added more detailed discussions in our revised manuscript.

Page 11, lines 8-14.

We also performed poststructure analysis for Co-N-CTS catalysts after the long-term stability test. The Co-N-CTS catalysts still maintained the open and hierarchical porous structure with high-density Co NPs uniformly dispersed on the 2D porous carbon, and no obvious agglomeration was found (**Figure S35**). Additionally, the XRD and XPS results showed a metallic Co phase with an increasing oxidized surface after the stability test of ORR and OER (**Figure S36**), which is commonly observed for nonnoble metals^{56–58}. These results validate the good structural and performance stability of the Co-N-CTS catalyst.

Modification in revised Supporting Information:

In addition, the XRD, XPS, SEM, and TEM results after the OER and ORR stability tests are also added to **Figure S31** and **Figure S34-36** of the Supporting Information as suggested by the reviewer.

3. Although the main point of this paper is introducing the formation of high-density NPs by combination of CTS and coordination chemistry, if there are some mechanistic studies of the ORR and OER, that would be very helpful and useful.

Response: We truly appreciate the reviewer’s valuable comment about the mechanistic studies of the ORR and OER. There are some very important works mainly focused on the mechanism study/review of ORR and OER, particularly using the Co catalysts (*Adv. Mater.* 2018, 30, 1705431, *Adv. Mater.* 2019, 31, 1901666, *Adv. Funct. Mater.* 2023, 33, 2209726). Although the catalytic process on our Co-N-CTS samples is similar to these important references, our catalyst demonstrated outstanding electrochemical performance (**Figure R33**).

Figure R33. Comparison of the bifunctional oxygen electrocatalytic performances ($E_{1/2}$ in ORR and $E_{j=10}$ in OER). The marked data are the potential difference between OER ($E_{j=10}$) and ORR ($E_{1/2}$) ($\Delta E = E_{j=10} - E_{1/2}$) between Co-CTS and other reported Co-based bifunctional electrocatalysts in the literature.

Structure-property relationship: The superior catalytic activity (and stability) of our Co-N-CTS catalysts largely benefited from the unique structures prepared by the coordinated CTS process. The major differences come from the fact that our Co-N-CTS catalysts not only have an ultrasmall size and high-density dispersion, which increases the active surface area (more active sites) but also exhibit open and hierarchical porous structures (micro-sized cavities and pores between carbon layers), which significantly benefit catalyst exposure, electrolyte penetration, and gas transport, especially at high overpotentials. Importantly, we confirmed the strong metal-support interaction and Co-N bond in our samples with clear electron/charge transfer. Co-N coordination as well as electron transfer can largely tailor the electronic structure of catalysts and affect their intrinsic catalytic activity. The strong Co-N bonds and coordination are also helpful to improve catalyst stability by inducing the strong metal-substrate interaction (SMSI) and even an encapsulate overlayer to chemically and physically stabilize the Co nanoparticles.

Below is a brief discussion on the structure-property relationship in Co-N-CTS (**Figure R34**).

First, the strong metal-substrate interaction between Co NPs and N-doped carbon and their electron transfer significantly regulated the electronic structure of Co to optimize the binding energies toward intrinsically enhanced catalytic activity (*Angew. Chemie Int. Ed.* 2019, 58, 12540-12544, *ACS Catal.* 2020, 10, 12376-12384). Second, the high-density and ultrasmall Co NPs with more surface and low-coordinated atoms afford abundant active sites to promote an increase in the apparent catalytic activity (*Adv. Funct. Mater.* 2020, 30, 1906081, *Adv. Energy Mater.* 2019, 9, 1900149). Third, the catalysts are dispersed on open and hierarchical porous structures with large surface areas, which effectively exposes these active sites, facilitates electrolyte penetration, and benefits gas/mass transportation, all of which are extremely beneficial for electrocatalytic ORR and OER, especially at large current densities.

Figure R34. Schematic image of Co-N-CTS for OER and ORR electrocatalysis.

Modification in the revised manuscript:

Following the reviewer's valuable suggestion, to help the understanding of the ORR and OER, we added more detailed discussions in our revised manuscript.

Page 11, lines 15-23; page 12, line 1 and 2.

The outstanding ORR and OER activity of our catalyst is comparable or superior to that of most of the reported state-of-the-art nonprecious metal catalysts (**Figure S37** and **Table S5**),

which could be attributed to the following reasons. For one, the high-density and ultrasmall Co NPs with more surface and low-coordinated atoms afford abundant active sites to promote and increase the apparent catalytic activity^{59,60}. In addition, the strong metal-substrate interaction between Co NPs and N-doped carbon and their electron transfer significantly regulate the electronic structure of Co to optimize the binding energies toward intrinsically enhanced catalytic activity^{41,61}. Last but not least, the catalysts are dispersed on open and hierarchical porous structures with large surface areas, which effectively exposes these active sites, facilitates electrolyte penetration, and benefits gas/mass transportation, all of which are extremely beneficial for electrocatalytic ORR and OER, especially at large current densities.

REVIEWER COMMENTS

Reviewer #1 (Remarks to the Author):

The work is now suitable for publication.

Reviewer #2 (Remarks to the Author):

The authors have addressed most of my concerns. I believe this work can be accepted by Nat Commun after the below remaining issues are addressed:

1. About the comparison of previous CTS with the reported CTS here, the authors said “rapid CTS pyrolysis was used for MOF microcubes to synthesize ultrasmall NPs (~ 4 nm) , yet most NPs were found to be buried inside the carbon matrix”. I am little confused here, aren't the nanoparticles supported on the surface of the carbon support during the CTS synthesis in the Science 2018 and later papers?
2. I still have questions on the control of the heating rate. In the rebuttal letter, the authors mentioned the heating rates (100, 1000, 10^4 °C/s) could be controlled by the current input. The precise heating rate control is hard to achieve since it depends not only the current input but also the sample resistance, heat capacity, thermal dissipation, etc. Could the author please provide the current profiles and temperature profiles of the sample during the heating?
3. About the chemical state of Co species, XPS shows the existence of Co(0), Co²⁺, Co³⁺ in both Co-CTS and Co-N-CTS, yet the EXAFS suggest only Co(0) in Co-CTS. Explain the inconsistency between these two characterizations.

Point-by-point responses to reviewers' remarks

(Black italic: Reviewer's comments; Blue type: Our response; Red type: Our revised)

Reviewer #1:

The work is now suitable for publication.

Response: We appreciate the reviewer for his/her recommending our manuscript for publication in Nature Communications.

Reviewer #2 (Remarks to the Author):

The authors have addressed most of my concerns. I believe this work can be accepted by Nat Commun after the below remaining issues are addressed:

Response: We sincerely appreciate the reviewer for the previous valuable comments which have made our manuscript stronger.

1. About the comparison of previous CTS with the reported CTS here, the authors said "rapid CTS pyrolysis was used for MOF microcubes to synthesize ultrasmall NPs (~ 4 nm), yet most NPs were found to be buried inside the carbon matrix". I am little confused here, aren't the nanoparticles supported on the surface of the carbon support during the CTS synthesis in the Science 2018 and later papers?

Response: We are grateful for the reviewer's valuable comment about the distribution of nanoparticles. First, we agree with the reviewer that the nanoparticles are supported on the surface of carbon support during the CTS synthesis in the Science 2018 and later papers, as shown in **Figure R1**.

Figure R1. (a) SEM images of synthesized AuNi nanoparticles on CNFs carbonized at different temperatures. (b) SEM images of Cu, Au, and Pt particle distributions synthesized on identical CNF supports via the same CTS process. (*Science*. 2018. 359:1489-1494.)

In the rapid pyrolysis of MOFs by CTS, the MOF materials are also dispersed on the carbon support, similar as previous nanoparticles. However, since MOF materials contain both metal

nodes and ligands, after CTS pyrolysis, those ligands will be pyrolyzed into carbon frameworks, resulting in the metal nanoparticles buried in these carbon frameworks. Namely, the nanoparticles are buried inside the carbon frameworks derived from pyrolyzing MOF ligands, and they are not buried inside the carbon support used for CTS. As shown in **Figure R2a and 2b**, the MOF microcubes derivatives are still supported on the surface of carbon support (carbon paper) after rapid CTS pyrolysis. However, the nanoparticles are buried inside the carbon matrix derived from pyrolyzing MOF ligands (**Figure R2c and 2d**).

Figure R2. (a, b) SEM images of MOF microcubes derivatives supported on carbon cloth. (c) SEM image of MOF microcubes derivatives. (d) TEM image of MOF microcubes derivatives. (*Nano Energy* 97 (2022) 107125)

Modification in the revised manuscript:

Following the reviewer’s suggestion, we revised the description in our revised manuscript. Page 4, lines 6 and line 7.

yet most NPs were found to be buried inside the carbon matrix derived from pyrolyzing MOF ligands.

2. I still have questions on the control of the heating rate. In the rebuttal letter, the authors mentioned the heating rates (100, 1000, 10^4 °C/s) could be controlled by the current input. The precise heating rate control is hard to achieve since it depends not only the current input but also the sample resistance, heat capacity, thermal dissipation, etc. Could the author please provide the current profiles and temperature profiles of the sample during the heating?

Response: We appreciate the reviewer’s comment on the heating rate controlled by the CTS

process. We agree with the reviewer that precise heating rate control (also temperature control) is difficult as it depends on many factor. Fortunately, these parameters are fixed for highly graphitized carbon supports and can be found by tuning the Joule heating parameters.

Below we demonstrate an example of the temperature and heating rates control by programming the electrical current. The carbon paper slice (length * width * thickness = 3.0 cm * 0.5 cm * 0.03 mm) is used as the heater. We adopt the current output mode and the maximum output voltage is 30 V. In our heating rate experiments, a heating rate of $\sim 10^4$ °C/s is achieved by an instant pulsing current to 10 A in 100 ms (~ 100 A/s) (**Figure R3a, Table R1**). Other slower heating rates (100 and 1000 °C/s) can be achieved by programming the current stepwise at rates of 1 A/s and 10 A/s from 0 to 10 A (**Figure R3b and 3c, Table R1**). From the temperature analysis, the heating rates are 106.3, 1042, 10090 °C/s for different current stepwise input (1, 10, 100 A/s) from 0 to 10 A, respectively, which is very close to our design and can be used to study the rate effect in the coordinated CTS process.

Figure R3. The current and temperature profiles at different (designed) heating rates: (a) 10^4 °C/s, (b) 1000 °C/s, and (c) 100 °C/s.

Table R1. The heating rates control at different target heating rate.

Target heating rate (°C/s)	Current rate (A/s)	Current range (A)	Heating time (s)	Peak temperature (°C)	Actual heating rate (°C/s)
100	1	0 ~ 10	10	1063	106.3
1000	10	0 ~ 10	1	1042	1042
10000	100	0 ~ 10	0.1	1009	10090

Note: the heater size (carbon paper): length * width * thickness = 3.0 cm * 0.5 cm * 0.03 mm; Power loading mode: current output; Maximum output voltage: 30 V.

Modification in the revised manuscript:

Following the reviewer's suggestion, to help the reader's understanding of the control of the heating rate, we added more detailed discussions in our revised manuscript.

Page 18, lines 18-22.

The carbon paper slice (length * width * thickness = 3.0 cm * 0.5 cm * 0.03 mm) is used as the heater. The current output mode is adopted and the maximum output voltage is 30 V. The fast heating rate of $\sim 10^4$ °C/s is achieved by an instant pulsing current to 10 A in 100 ms (~ 100 A/s), and other slower heating rates (100 and 1000 °C/s) can be achieved by

programming the current stepwise at rates of 1 A/s and 10 A/s from 0 to 10 A.

Modification in revised Supporting Information:

In addition, the current and temperature profiles are also added as **Figure S9** and **Table S2** in Supporting Information as suggested by the reviewer.

3. About the chemical state of Co species, XPS shows the existence of Co(0), Co²⁺, Co³⁺ in both Co-CTS and Co-N-CTS, yet the EXAFS suggest only Co(0) in Co-CTS. Explain the inconsistency between these two characterizations.

Response: We thank the reviewer for the comment. The major differences come from the fact that XPS is more sensitive to surface chemistry while EXAFS has more bulk information. Since the Co nanoparticles formed in Co-CTS is only slightly surface oxidized, this can be obviously captured by XPS, while the oxidation signal (Co-O bond) in EXAFS will become weak because of bulk averaging.

It should be noted that EXAFS did not show only metallic Co in Co-CTS but also slight surface oxidation signals. As shown in **Figure R4a**, the absorption near edge spectrum of Co-CTS lies near that for Co foil and shifts a little bit to the right compared Co foil, indicating the coexistence of metallic Co and a small amount of surface oxidized Co. Additionally, the Fourier transformed EXAFS spectra of the Co K-edge for Co-CTS shows a predominant peak at ≈ 2.16 Å and a small peaks at ≈ 1.48 Å (**Figure R4b**), which is typically assigned to the Co-Co and Co-O coordination, respectively. Therefore, the inconsistency between these two characterizations comes from that (1) Co nanoparticle in Co-CTS sample is only slightly surface oxidized and (2) the XPS mainly investigated the surface signal of samples (more Co²⁺ and Co³⁺ signal), while the EXAFS provides more bulk information of samples (relatively weak Co-O information) because of bulk averaging.

Figure R4. (a) Normalized XANES spectra at the Co K-edge of Co-N-CTS, Co-CTS, and reference samples. The inset shows the enlarged marked area. (b) k^3 -weighted Fourier transform of EXAFS spectra at the Co K-edge of Co-N-CTS, Co-CTS, and reference samples.

Modification in the revised manuscript:

Following the reviewer's suggestion, to help the reader's understanding of the chemical state of Co species, we added more detailed discussions in our revised manuscript.

Page 9, line 16-18.

the absorption near the edge spectrum of Co-CTS lies near that for Co foil and shifts slightly to the right, indicating the coexistence of metallic Co and slight surface oxidized Co.

Page 9, line 20-22.

whereas for Co-CTS, only a predominant peak at $\approx 2.16 \text{ \AA}$ and a weak peaks at $\approx 1.48 \text{ \AA}$ were observed, which is typically assigned to the Co-Co and Co-O coordination, respectively⁵³⁻⁵⁵.